# Synthesis of novel *N*-substituted *β*-amino acid derivatives bearing 2-hydroxyphenyl moieties as promising antimicrobial candidates targeting multidrug-resistant Gram-positive pathogens

**Povilas Kavaliauskas**[1,2,3,4*], **Birutė Grybaitė**[1], **Birute Sapijanskaite-Banevič**[1],
**Rūta Petraitien**ė[2,3,5], **Ramun**ė **Grigalevičiūt**ė[4,6], **Andrew Garcia**[2], **Ethan Naing**[2],
**Vytautas Mickevičius**[1], **Sergey Belyakov**[7], **Vidmantas Petraitis**[2,3,5]

1 Department of Organic Chemistry, Kaunas University of Technology, Kaunas, Lithuania, 2 Division of Infectious Diseases, Department of Medicine, Weill Cornell Medicine of Cornell University, New York, New York, United States of America, 3 Institute of Infectious Diseases and Pathogenic Microbiology, Prienai, Lithuania, 4 Biological Research Center, Lithuanian University of Health Sciences, Kaunas, Lithuania, 5 Center for Discovery and Innovation, Hackensack Meridian Health, Nutley, New Jersey, United States of America, 6 Department of Animal Nutrition, Lithuanian University of Health Sciences, Kaunas, Lithuania, 7 Latvian Institute of Organic Synthesis, Laboratory of Physical Organic Chemistry, Riga, Latvia

☙ These authors contributed equally to this work.
* povilas.kavaliauskas@som.umaryland.edu

## Abstract

The increasing prevalence of antimicrobial resistance among ESKAPE group pathogens presents a significant challenge in the healthcare sector, contributing to higher morbidity and mortality rates globally. Thus, it is essential to develop novel antimicrobial agents effective against drug-resistant pathogens. In this study, we report the synthesis and *in vitro* antimicrobial activity characterization of novel *N*-substituted *β*-amino acid derivatives bearing 2-hydroxyphenyl core against multidrug-resistant bacterial pathogens. The synthesized compounds (**2–26**) exhibited promising antimicrobial activity specifically against Gram-positive bacteria, with minimum inhibitory concentrations (MIC) ranging from 4 to 128 µg/mL. Compounds **9** (R = 4-nitrophenyl), **17** (R = 5-nitro-2-thienyl), **18** (R = 5-nitro-2-furyl), thiosemicarbazide **16**, and **26** exhibited the most promising activity against *Staphylococcus aureus* MRSA USA300 lineage strain TCH-1516, with MIC values between 4 and 16 µg/mL. Compound **26** demonstrated strong antimicrobial activity against both *S. aureus* TCH-1516 and *E. faecalis* AR-0781, with the activity comparable to control antibiotics. Furthermore, compound **26** exhibited antifungal activity drug-resistant against *Candida albicans* AR-0761 (MIC 16 µg/mL). These findings indicate that *N*-substituted *β*-amino acid derivatives with a 2-hydroxyphenyl core warrant further investigation as a potential scaffold for the further development of antimicrobial agents based on compound **26** targeting Gram-positive pathogens and drug-resistant *C. albicans* AR-0761.

**Data availability statement:** All relevant data are within the manuscript and its Supporting information files.

**Funding:** The author(s) received no specific funding for this work.

**Competing interests:** The authors have declared that no competing interests exist.

## Introduction

The growing prevalence of antimicrobial resistance represents a critical challenge in healthcare setting, with the increasing numbers of infections caused by ESKAPE group pathogens [1–3]. This group comprises of drug-resistant *Enterococcus faecium*, *Staphylococcus aureus*, *Klebsiella pneumoniae*, *Acinetobacter baumannii*, *Pseudomonas aeruginosa*, and *Enterobacter* species [4,5]. These pathogens are especially challenging for their antimicrobial resistance to multiple first line antibiotics, thereby complicating treatment regimens and leading to higher rates of morbidity and mortality [5]. The clinical impact of ESKAPE pathogens is profound, with methicillin-resistant *Staphylococcus aureus* (MRSA) and vancomycin-resistant *Enterococci* (VRE) being prominent contributors to severe healthcare-associated infections [4,6,7]. Among clinically important *Enterococcus, E. faecalis* remains important clinical pathogen often acquiring multiple resistance mechanisms therefore making a treatment of infections caused by *E. faecalis* often challenging [6,7]. The high mortality rates associated with these infections, coupled with the prolonged hospital stays and increased healthcare costs, underscore the urgent need for the development of novel antimicrobial agents for further pre-clinical development and optimization [8,9]. Infections caused by MRSA are known to cause a variety of infections, including skin and soft tissue infections, pneumonia, endocarditis, osteomyelitis, and sepsis, with mortality rates ranging from 20% to 50% depending on the severity and site of infection [10–12]. VRE predominantly causes bloodstream infections, urinary tract infections, intra-abdominal infections, and wound infections, with mortality rates for VRE bacteremia reaching up to 60% [13–15].

The molecular mechanisms by which these pathogens evade antibiotics are complex and multifaceted [16]. MRSA confers resistance to β-lactam antibiotics primarily through the acquisition of the *mecA* gene, which encodes a penicillin-binding protein (PBP2a) with low affinity for β-lactams, thereby preventing the inhibition of cell wall synthesis [16,17]. Additionally, MRSA employs various other resistance mechanisms, such as efflux pumps and the production of various antibiotics-modifying enzymes, often conferring resistance to aminoglycosides [18,19]. VRE, on the other hand, utilizes modification in cell wall precursors biosynthesis to evade vancomycin-mediated killing [20]. The acquisition of *vanA* or *vanB* gene clusters results in the modification of the terminal D-alanine-D-alanine to D-alanine-D-lactate in the peptidoglycan precursor, significantly reducing vancomycin binding affinity in VRE cell wall biogenesis pathway [20–22]. This modification impedes the antibiotic's ability to inhibit cell wall synthesis, thereby conferring resistance to vancomycin [20]. The clinical impact of these resistance mechanisms is profound, as they severely limit the therapeutic options available for treating MRSA and VRE infections, necessitating the development of new antimicrobial strategies [21,22].

Various amino acid derivatives have been extensively investigated in medicinal chemistry as novel pharmacophores for targeting a variety of diseases, including cancers, bacterial and fungal infections, and parasitic diseases (Fig 1) [22–26].

These derivatives are attractive candidates for drug design due to their inherent biological activity, structural diversity, and ability to mimic natural substrates

**Fig 1. Pharmaceuticals containing β-amino acid structures.**

in biological systems [23,25,26]. For instance, modifications of amino acid side chains or addition of various aromatic or heterocyclic substituents can enhance binding affinity to specific biological targets, improve selectivity, and increase metabolic stability [22,23,26]. Especially 1,3,4-oxadiazole-2(3*H*)-thione fragment are an important part of heterocyclic compounds with broad spectrum of biological activities. Substituted 1,3,4-oxadiazoles have revealed antibacterial, anti-mycobacterial, antifungal, anti-inflammatory, analgesic, anticonvulsant and anticancer properties [24–26]. Among substituted amino acid derivatives, *N*-substitution in β-amino acids offers promising scaffold with remarkable versatility in tailoring molecular properties. This *N*-substituted scaffold allows to incorporate a wide variety of substitutions that are crucial for further optimization of physicochemical attributes, influencing lipophilicity, electronic distribution, and overall bioactivity [22–27].

Our previous studies have demonstrated that amino acid derivatives bearing a 4-hydroxyphenyl group exhibit promising antimicrobial activity against multidrug-resistant bacterial pathogens [28]. Additionally, the synthesized compounds exhibited significant antifungal activity against highly multidrug-resistant fungal pathogens, including the emerging azole-resistant *Candida auris* [28]. Further in silico analyses indicated favorable predicted pharmacological and drug-like properties, establishing 3-((4-hydroxyphenyl)amino)propanoic acid as a promising scaffold for the development of antimicrobial candidates [28]. Building on these findings, the present study explores the synthesis pathways and *in vitro* antimicrobial properties of new *N*-substituted β-amino acid derivatives bearing a 2-hydroxyphenyl moiety, along with various aromatic and heterocyclic substituents. In this study, we successfully demonstrate that *N*-substituted β-amino acid derivatives with a 2-hydroxyphenyl core, as well as heterocyclic substituents, could serve as promising scaffolds for the development of novel antimicrobial candidates targeting Gram-positive pathogens.

## Materials and methods

### Chemical synthesis

**General analytical procedures.** The reaction course and purity of the synthesized compounds were monitored by TLC using aluminum plates precoated with Silica gel with F254 nm (Merck KGaA, Darmstadt, Germany). Reagents and solvents were obtained from Sigma-Aldrich (St. Louis, MO, USA) and used without further purification. Melting points were determined with a B-540 melting point analyzer (Büchi Corporation, New Castle, DE, USA) and were uncorrected. IR spectra (υ, cm$^{-1}$) were recorded on a Perkin–Elmer Spectrum BX FT–IR spectrometer using KBr pellets. NMR spectra were recorded on a Brucker Avance III (400, 101 MHz) spectrometer. Chemical shifts were reported in (δ) ppm relative to tetramethylsilane (TMS) with the residual solvent as internal reference ([D$_6$]DMSO, δ = 2.50 ppm for $^1$H and δ = 39.5 ppm for $^{13}$C). $^{19}$F NMR spectra (376 MHz, absolute referencing via the Ξ ratio) were obtained on a Bruker Avance III 400 instrument with a ′directly′ detecting broadband observe probe (BBO). Data were reported as follows: chemical shift, multiplicity, coupling constant (Hz), integration, and assignment. Elemental analyses (C, H, N) were conducted using the Elemental Analyzer CE-440; their results were found to be in good agreement (±0.3%) with the calculated values.

**General procedures for preparation of compounds 2–26.** *3,3'-((2-Hydroxyphenyl)azanediyl)dipropionic acid (2)*: A mixture of *o*-aminophenol (**1**) (10.9 g, 100 mmol), acrylic acid (18 g, 250 mmol) and water (100 mL) was heated under reflux for 14 h and cooled. Crystaline product 2 was filtered off, washed with propan-2-ol, and dried. White powder, yield 18.97 g (75%), m.p. 181−183 °C (from propan-2-ol). $^1$H NMR (400 MHz, DMSO-*d$_6$*) δ: 2.29 (t, *J* = 7.1 Hz, 4H, 2x CH$_2$CO), 3.18 (t, *J* = 7.1 Hz, 4H, 2x NCH$_2$), 6.66–7.06 (m, 4H, H$_{Ar}$); 8.77 (br.s, 1H, OH); 12.06 (br.s, 2H, 2x OH); $^{13}$C NMR (101 MHz, DMSO-*d$_6$*) δ: 32.4 (CH$_2$CO), 48.6 (NCH$_2$), 115.6, 119.4, 123.1, 124.9, 136.5, 152.7 (C$_{Ar}$), 173.8 (2x C=O). IR (KBr): v$_{max}$ (cm$^{-1}$) = 3045, 2971 (3x OH), 1698 (2x C=O). Anal. Calcd. for C$_{12}$H$_{15}$NO$_5$, %: C 56.91; H 5.97; N 5.53. Found: C 56.69; H 5.70; N 5.33.

*3-(2-Oxo-3,4-dihydrobenzo[b][1,4]oxazepin-5(2H)-yl)-N-(4-sulfamoylphenyl)propenamide (3)*: The dipropionic acid **2** (0.5 g 2, mmol) were dissolved in dimethylformamide (2 mL) by heating, then sulfanilamide (0.83 g, 4.8 mmol,) were dissolved in dimethylformamide (2 mL) in another beaker and both prepared solutions were poured into a flask and triethylamine (12 mmol) were added dropwise, the reaction mixture was stirred 20 minutes at room temperature. Separately, HBTU (2.28 g, 6 mmol) were dissolved in 9 mL of dimethylformamide and were added dropwise to the reaction mixture. The reaction was stirred at room temperature for 24 h. After that, the reaction mixture were diluted with water (40 mL), the formed crystals were filtered off and were washed with 5% potassium carbonate solution, water and dried. White powder, yield 0.22 g (28%), m.p. 190−192 °C (from dioxane/water mixture). $^1$H NMR (400 MHz, DMSO-*d$_6$*) δ: 2.58 (t, *J* = 7.0 Hz, 4H, 2x CH$_2$CO), 3.39 (t, *J* = 7.0 Hz, 4H, 2x NCH$_2$), 6.99–7.16 (m, 2H, SO$_2$NH$_2$), 7.19–7.32 (m, 4H, H$_{Ar}$), 7.72 (q, *J* = 8.7 Hz, 4H, H$_{Ar}$), 12.21 (s, 1H, NH); $^{13}$C NMR (101 MHz, DMSO-*d$_6$*) δ: 31.7 (CH$_2$CONH), 35.0 (CH$_2$CONHO), 47.5 (NCH$_2$ CH$_2$CONH), 54.3 (NCH$_2$CH$_2$CONHO), 118.6, 119.5, 121.1, 123.9, 126.3, 126.6, 138.2, 138.4, 141.9, 146.9 (C$_{Ar}$), 169.9, 170.0 (2x C=O). IR (KBr): v$_{max}$ (cm$^{-1}$) = 3344 (NH$_2$), 3241 (NH), 1742, 1689 (2x C=O). Anal. Calcd. for C$_{18}$H$_{19}$N$_3$O$_5$S, %: C 55.52; H 4.92; N 10.79. Found: C 55.31; H 4.72; N 10.55. HRMS m/z calculated for C$_{18}$H$_{19}$N$_3$O$_5$S [M+H]+ : 390.1045; found: 390.1119.

*Dimethyl 3,3'-((2-hydroxyphenyl)azanediyl)dipropionate (4)*: A mixture of dipropionic acid **2** (17.48 g, 69 mmol), conc. sulfuric acid (8.6 g, 4.7 mL, 87 mmol) and methanol (250 mL) was heated under reflux for 7 h. Then the solvent was evaporated under reduced pressure, and the residue was neutralized with 5% sodium carbonate solution to pH 7. The obtained solid was filtered off, washed with plenty of water and recrystallized from propan-2-ol. Light brown powder, yield 15.7 g (81%), m.p. 87–89 °C. $^1$H NMR (400 MHz, DMSO-*d$_6$*) δ: 2.37 (t, *J* = 7.0 Hz, 4H, 2x CH$_2$CO), 3.22 (t, *J* = 7.0 Hz, 4H, 2x NCH$_2$), 3.54 (s, 6H, 2x CH$_3$), 6.68–7.05 (m, 4H, H$_{Ar}$); 8.66 (s, 1H, OH); $^{13}$C NMR (101 MHz, DMSO-*d$_6$*) δ: 32.9 (CH$_2$CO), 48.4 (NCH$_2$), 51.2 (CH$_3$), 115.5, 119.2, 123.3, 124.8, 135.9, 152.6 (C$_{Ar}$), 172.4 (2x C=O). IR (KBr): v$_{max}$ (cm$^{-1}$) = 3308 (OH), 1727 (2x C=O). Anal. Calcd. for C$_{14}$H$_{19}$NO$_5$, %: C 59.78; H 6.81; N 4.98. Found: C 59.52; H 6.60; N 4.72.

*3,3'-((2-Hydroxyphenyl)azanediyl)di(propanehydrazide) (5)*: A mixture of methyl ester **4** (7.28 g, 26 mmol), hydrazine hydrate (7.89 g, 157 mmol), and propan-2-ol (25 mL) was heated under reflux for 5 h and cooled. Crystalline product 5 was filtered off, washed with propan-2-ol, diethyl ether, and dried. White powder, yield 5.82 g (80%), m.p. 149−151 °C (from propan-2-ol). $^1$H NMR (400 MHz, DMSO-*d$_6$*) δ: 2.13 (t, *J* = 7.1 Hz, 4H, 2x CH$_2$CO), 3.12 (t, *J* = 7.2 Hz, 4H, 2x NCH$_2$), 4.20 (s, 4H, 2x NH$_2$), 6.70–7.08 (m, 4H, H$_{Ar}$); 9.01 (s, 3H, OH, 2x NHNH$_2$); $^{13}$C NMR (101 MHz, DMSO-*d$_6$*) δ: 31.6 (CH$_2$CO), 49.0 (NCH$_2$), 115.7, 119.1, 122.6, 124.6, 136.7, 152.8 (C$_{Ar}$), 170.6 (2x C=O). IR (KBr): v$_{max}$ (cm$^{-1}$) = 3295 (OH), 3178 (NH$_2$), 3057 (NH$_2$), 1677, 1624 (C=O). Anal. Calcd. for C$_{12}$H$_{19}$N$_5$O$_3$, %: C 51.23; H 6.81; N 24.90. Found: C 51.01; H 6.62; N 24.69.

*2-((2-(1H-benzo[d]imidazol-2-yl)ethyl)amino)phenol (6)*: Mixture of compound **2** (0.7 g, 0.0028 mol) and *o*-phenylenediamine (0.61 g, 0.0056 mol) in dilute hydrochloric acid (1:1, 8 mL) was heated at reflux for 24 h, then cooled down. The reaction mixture was neutralized with a 5% sodium carbonate solution to pH 8. The formed crystals was recrystallized from a mixture of 1,4-dioxane and water (1:1, 15 ml). Brown powder, yield 0.2 g (28%), m.p. 196−198 °C. $^1$H NMR (400 MHz, DMSO-*d$_6$*) δ: 3.10 (t, *J* = 7.0 Hz, 2H, 2x CH$_2$CN), 3.52 (q, *J* = 7.0 Hz, 2H, 2x NCH$_2$), 4.86 (t, *J* = 7.0 Hz, 1H, NHCH$_2$), 6.23–6.74 (m, 4H, H$_{Ar}$); 7.01–7.22 (m, 2H, H$_{Ar}$), 7.29–7.68 (m, 2H, H$_{Ar}$), 9.23 (s, 1H, OH), 12.28 (s, 1H, NH); $^{13}$C

NMR (101 MHz, DMSO-$d_6$) δ: 28.5 (<u>C</u>H$_2$CO), 41.3 (NCH$_2$), 109.8, 110.8, 113.5, 116.0, 118.1, 119.8, 120.9, 121.5, 134.2, 137.1, 143.3, 144.2, 153.4 (C$_{Ar}$). IR (KBr): ν$_{max}$ (cm$^{-1}$) = 3310 (OH), 3055 (2x NH). Anal. Calcd. for C$_{15}$H$_{15}$N$_3$O, %: C 71.13; H 5.97; N 16.59. Found: C 70.95; H 5.70; N 16.32.

*General procedure for the preparation of hydrazones* **7–22**: To a solution of hydrazide **5** (0.42g, 1.5 mmol) in 2-propanol (15 mL), the corresponding aromatic aldehyde was added (1.65 mmol) and the mixture was heated at reflux for 2h, then cooled down, and the formed precipitate was filtered off, washed with methanol, diethyl ether and recrystallizing from 1,4-dioxane.

*3,3'-((2-Hydroxyphenyl)azanediyl)bis(N'-(benzylidene)propanehydrazide)* (**7**): White powder, yield 0.57 g (83%), m.p. 223−225 °C. $^1$H NMR (400 MHz, DMSO-$d_6$) δ: 2.34 and 2.77 (2t, *J*=7.1 Hz, 2H, 2x CH$_2$CO), 3.17–3.41 (m, 4H, 2x NCH$_2$), 6.68–7.21 (m, 4H, H$_{Ar}$); 7.29–7.79 (m, 10H, H$_{Ar}$), 7.83–8.19 (m, 2H, 2x CH), 8.91, 8.96, 9.02 (3s, 1H, OH), 11.30, 11.31, 11.41 (3s, 2H, 2x NH); $^{13}$C NMR (101 MHz, DMSO-$d_6$) δ: 30.2, 30.4, 31.6, 32.4, 32.6 (<u>C</u>H$_2$CO), 48.3, 48.5, 48.7, 48.9 (NCH$_2$), 115.6, 119.1, 119.2, 122.4, 122.8, 124.3, 124.6, 126.6, 126.7, 126.9, 128.7, 128.8, 129.7, 129.9, 134.2, 134.3, 136.2, 136.5, 136.9, 152.8, 145.9, 152.4, 152.5, 152.8, 167.5, 167.6 (C$_{Ar}$) 170.6, 173.4, 173.5 (C=O). IR (KBr): ν$_{max}$ (cm$^{-1}$) = 3109 (OH), 3007 (2x NH), 1672 (2x C=O). Anal. Calcd. for C$_{26}$H$_{27}$N$_5$O$_3$, %: C 68.25; H 5.95; N 15.31. Found: C 68.00; H 5.73; N 15.11. HRMS m/z calculated for C$_{26}$H$_{27}$N$_5$O$_3$ [M+H]+ : 458.2114; found: 458.2184.

*3,3'-((2-Hydroxyphenyl)azanediyl)bis(N'-(2,4-difluorobenzylidene)propanehydrazide)* (**8**): White powder, yield 0.57 g (83%), m.p. 224−226 °C. $^1$H NMR (400 MHz, DMSO-$d_6$) δ: 2.23–2.39 and 2.64–2.87 (2m, 4H, 2x CH$_2$CO), 3.15–3.34 (m, 4H, 2x NCH$_2$), 6.68–7.37 (m, 8H, H$_{Ar}$); 7.58–7.95 (m, 2H, H$_{Ar}$), 7.99–8.11 and 8.20–8.30 (2m, 2H, 2x N=CH), 8.88, 8.90, 9.00 (3s, 1H, OH), 11.39, 11.41, 11.53 (3s, 2H, 2x NH); $^{13}$C NMR (101 MHz, DMSO-$d_6$) δ: 30.1, 30.4, 32.4, 32.6 (<u>C</u>H$_2$CO), 48.2, 48.4, 48.6, 48.7 (NCH$_2$), 104.1, 104.4, 104.6, 112.5, 115.6, 118.5, 118.6, 119.01, 119.2, 122.3, 122.7, 124.4, 124.6, 127.5, 127.8, 127.9, 134.8, 136.5, 136.8, 137.9, 152.4, 152.5, 159.3, 159.5, 161.6, 161.9, 164.1, 167.6, 167.7 (C$_{Ar}$), 173.5, 173.6 (2x C=O). $^{19}$F NMR (376 MHz, DMSO-$d_6$) δ: −107.2, −107.5, −117.0, −117.3. IR (KBr): ν$_{max}$ (cm$^{-1}$) = 3297 (OH), 3196 (2x NH), 1673 (2x C=O). Anal. Calcd. for C$_{26}$H$_{23}$F$_4$N$_5$O$_3$, %: C 58.98; H 4.38; N 13.23. Found: C 58.73; H 4.13; N 13.03. HRMS m/z calculated for C$_{26}$H$_{23}$F$_4$N$_5$O$_3$ [M+H]+ : 530.1737; found: 530.1810.

*3,3'-((2-Hydroxyphenyl)azanediyl)bis(N'-(4-nitrobenzylidene)propanehydrazide)* (**9**): Light yellow powder, yield 0.55 g (67%), m.p. 225−227 °C (from propan-2-ol). $^1$H NMR (400 MHz, DMSO-$d_6$) δ: 2.38 (t, *J*=6.8 Hz, 2H, CH$_2$CO), 2.80 (t, *J*=6.7 Hz, 2H, CH$_2$CO), 3.25−3.39 (m, 4H, 2x NCH$_2$), 6.73–8.30 (m, 14H, H$_{Ar}$, 2x N=CH); 8.90, 8.93, 9.06 (3s, 1H, OH), 11.58, 11.60, 11.71 (3s, 2H, 2x NH); $^{13}$C NMR (101 MHz, DMSO-$d_6$) δ: 30.3, 30.5, 32.6 (<u>C</u>H$_2$CO), 48.1, 48.4, 48.7, 48.8 (NCH$_2$), 115.6, 115.8, 119.1, 119.3, 119.4, 122.3, 122.8, 123.1, 123.9, 124.3, 124.6, 127.4, 127.5, 127.8, 136.2, 136.5, 136.8, 140.4, 140.5, 140.5, 140.7, 143.8, 143.6, 147.4, 147.5, 147.6, 147.7, 152.3, 152.5, 152.8, 168.0 (C$_{Ar}$), 173.8, 173.9 (2x C=O). IR (KBr): ν$_{max}$ (cm$^{-1}$) = 3265 (OH), 3092 (2x NH), 1675 (2x C=O). Anal. Calcd. for C$_{26}$H$_{25}$N$_7$O$_7$, %: C 57.04; H 4.60; N 17.91. Found: C 56.83; H 4.42; N 17.75.

*3,3'-((2-Hydroxyphenyl)azanediyl)bis(N'-(4-chlorobenzylidene)propanehydrazide)* (**10**): White powder, yield 0.55 g (69%), m.p. 220−222 °C (from dioxane). $^1$H NMR (400 MHz, DMSO-$d_6$) δ: 2.34 (t, *J*=6.8 Hz, 2H, CH$_2$CO), 2.76 (t, *J*=7.1 Hz, 2H, CH$_2$CO), 3.25−3.36 (m, 4H, 2x NCH$_2$), 6.70–8.12 (m, 14H, H$_{Ar}$, 2x N=CH); 8.89, 8.94, 9.02 (3s, 1H, OH), 11.35, 11.36, 11.47 (3s, 2H, 2x NH); $^{13}$C NMR (101 MHz, DMSO-$d_6$) δ: 30.2, 30.4, 32.4, 32.6 (<u>C</u>H$_2$CO), 48.2, 48.4, 48.7, 48.9 (NCH$_2$), 115.6, 115.7, 119.1, 119.2, 119.3, 122.4, 122.7, 123.1, 124.3, 124.6, 124.9, 128.2, 128.3, 128.6, 128.8, 133.1, 133.3, 134.1, 134.3, 136.2, 136.5, 136.9, 144.7, 144.8, 152.3, 152.5, 152.8, 167.6, 167.7 (C$_{Ar}$), 173.5 (2x C=O). IR (KBr): ν$_{max}$ (cm$^{-1}$) = 3179 (OH), 3107 (2x NH), 1673 (2x C=O). Anal. Calcd. for C$_{26}$H$_{25}$Cl$_2$N$_5$O$_3$, %: C 59.32; H 4.79; N 13.30. Found: C 59.11; H 4.54; N 13.17. HRMS m/z calculated for C$_{26}$H$_{25}$Cl$_2$N$_5$O$_3$ [M+Na]+ : 548.1334; found: 548.1228.

*3,3'-((2-Hydroxyphenyl)azanediyl)bis(N'-(4-(dimethylamino)benzylidene)propanehydrazide)* (**11**): White powder, yield 0.75 g (91%), m.p. 206−208 °C. $^1$H NMR (400 MHz, DMSO-$d_6$) δ: 2.30 and 2.73 (2q, *J*=6.5 Hz, 2H, 2x CH$_2$CO), 2.91, 2.93, 2.95 (3s, 12H, 4x CH$_3$); 3.20–3.33 (m, 4H, 2x NCH$_2$), 6.56–6.98 (m, 6H, H$_{Ar}$); 7.06–7.17 (m, 1H, H$_{Ar}$), 7.29–7.51 (m, 4H, H$_{Ar}$), 7.76–8.01 (m, 2H, 2x N=CH), 9.00, 9.02, 9.06 (3s, 1H, OH), 11.03, 11.12, 11.13 (3s, 2H, 2x NH); $^{13}$C NMR (101

MHz, DMSO-$d_6$) δ: 30.3, 30.5, 32.3, 32.4, 32.5 (4x CH$_3$, $\underline{C}$H$_2$CO), 48.2, 48.6, 49.0 (NCH$_2$), 111.7, 111.8, 115.6, 119.2, 121.6, 122.2, 122.6, 123.01, 124.3, 124.5, 127.9, 128.3, 128.4, 136.3, 136.7, 137.2, 143.8, 146.9, 147.0, 151.2, 151.4, 152.5, 166.9, 167.0 (C$_{Ar}$), 172.9 (2x C=O). IR (KBr): ν$_{max}$ (cm$^{-1}$) = 3437 (OH), 3078 (2x NH), 1661 (2x C=O). Anal. Calcd. for C$_{30}$H$_{37}$N$_7$O$_3$, %: C 66.28; H 6.86; N 18.03. Found: C 66.04; H 6.65; N 17.86. HRMS m/z calculated for C$_{30}$H$_{37}$N$_7$O$_3$ [M+H]+ : 544.2957; found: 544.3025.

*3,3'-((2-Hydroxyphenyl)azanediyl)bis(N'-(4-hydroxybenzylidene)propanehydrazide)* (*12*): White powder, yield 0.45 g (62%), m.p. 164−166 °C (from methanol). $^1$H NMR (400 MHz, DMSO-$d_6$) δ: 2.31 (t, $J$ = 6.9 Hz, 2H, CH$_2$CO), 2.74 (t, $J$ = 6.7 Hz, 2H, CH$_2$CO), 3.19−3.36 (m, 4H, 2x NCH$_2$), 6.69–8.07 (m, 14H, H$_{Ar}$, 2x N=CH); 8.98 (t, 1H, OH), 9.84, 9.87 (2s, 2H, 2x OH), 11.10, 11.21 (2s, 2H, 2x NH); $^{13}$C NMR (101 MHz, DMSO-$d_6$) δ: 30.2, 30.4, 32.3, 32.5 ($\underline{C}$H$_2$CO), 48.3, 48.6, 48.7, 49.0 (NCH$_2$), 115.6, 115.7, 119.1, 119.2, 119.3, 122.4, 122.7, 123.1, 125.2, 125.3, 128.4, 128.4, 128.8, 136.3, 136.6, 137.1, 143.2, 146.4, 146.5, 152.4, 152.6, 152.8, 159.1, 159.3, 167.2, 167.3 (C$_{Ar}$), 173.1, 173.2 (2x C=O). IR (KBr): ν$_{max}$ (cm$^{-1}$) = 3577 (OH), 3088 (2x NH), 1654 (2x C=O). Anal. Calcd. for C$_{26}$H$_{27}$N$_5$O$_5$, %: C 63.79; H 5.56; N 14.31. Found: C 63.55; H 5.36; N 14.15. HRMS m/z calculated for C$_{26}$H$_{27}$N$_5$O$_5$ [M+Na]+ : 512.2012; found: 512.1903.

*3,3'-((2-Hydroxyphenyl)azanediyl)bis(N'-(3,4,5-trimethoxybenzylidene)propanehydrazide)* (*13*): White powder, yield 0.8 g (83%), m.p. 214−216 °C (from methanol). $^1$H NMR (400 MHz, DMSO-$d_6$) δ: 2.33 and 2.77 (q, $J$ = 6.9 Hz, 4H, CH$_2$CO), 3.12−3.36 (m, 4H, 2x NCH$_2$), 3.60–3.85 (m, 18H, 6x OCH$_3$), 6.66–7.17 (m, 8H, H$_{Ar}$); 7.83, 7.87, 8.02, 8.04 (4s, 2H, 2x N=CH); 8.81, 8.92, 9.02 (3s, 1H, OH), 11.31, 11.34, 11.36, 11.39 (4s, 2H, 2x NH); $^{13}$C NMR (101 MHz, DMSO-$d_6$) δ: 30.2, 30.4, 32.4, 32.6 ($\underline{C}$H$_2$CO), 48.7, 48.9, 55.8, 55.9, 60.1 (NCH$_2$, 6x OCH$_3$), 103.7, 103.8, 104.2, 115.3, 115.5, 119.0, 119.2, 122.9, 124.7, 129.7, 129.8, 136.3, 136.5, 136.7, 138.8, 138.9, 139.0, 142.5, 142.6, 145.9, 146.0, 152.7, 153.1, 167.4, 167.5 (C$_{Ar}$), 173.4 (2x C=O). IR (KBr): ν$_{max}$ (cm$^{-1}$) = 3327 (OH), 3111 (2x NH), 1670 (2x C=O). Anal. Calcd. for C$_{32}$H$_{39}$N$_5$O$_9$, %: C 60.27; H 6.16; N 10.98. Found: C 60.03; H 5.94; N 10.71.

*3,3'-((2-Hydroxyphenyl)azanediyl)bis(N'-(naphthalen-1-ylmethylene)propanehydrazide)* (*14*): White powder, yield 0.73 g (87%), m.p. 213−215 °C (from methanol). $^1$H NMR (400 MHz, DMSO-$d_6$) δ: 2.42 and 2.86 (t, $J$ = 7.2 Hz, 4H, CH$_2$CO), 3.34−3.52 (m, 4H, 2x NCH$_2$), 6.71–6.99 (m, 3H, H$_{Ar}$); 7.11–7.25 (m, 1H, H$_{Ar}$); 7.44–8.05 (m, 12H, H$_{Ar}$); 8.47–8.85 (m, 4H, H$_{Ar}$, 2x N=CH); 8.94, 9.02, 9.10 (3s, 1H, OH), 11.35, 11.37, 11.53, 11.54 (4s, 2H, 2x NH); $^{13}$C NMR (101 MHz, DMSO-$d_6$) δ: 30.3, 30.6, 32.4, 32.6 ($\underline{C}$H$_2$CO), 48.5, 48.6, 48.8, 48.9 (NCH$_2$), 115.5, 115.6, 115.8, 119.2, 122.7, 122.9, 123.6, 124.3, 124.5, 124.7, 125.5, 126.1, 126.2, 127.0, 127.1, 127.2, 127.3, 127.9, 128.8, 129.3, 129.4, 129.5, 129.9, 130.1, 130.3, 133.4, 133.5, 136.5, 136.8, 142.6, 142.7, 145.9, 146.1, 152.6, 152.7, 167.5, 167.6 (C$_{Ar}$), 173.3, 173.4 (2x C=O). IR (KBr): ν$_{max}$ (cm$^{-1}$) = 3169 (OH), 3092 (2x NH), 1675 (2x C=O). Anal. Calcd. for C$_{34}$H$_{31}$N$_5$O$_3$, %: C 73.23; H 5.60; N 12.56. Found: C 73.02; H 5.43; N 12.33.

*3,3'-((2-Hydroxyphenyl)azanediyl)bis(N'-(furan-2-ylmethylene)propanehydrazide)* (*15*): White powder, yield 0.46 g (70%), m.p. 194−196 °C (from dioxane). $^1$H NMR (400 MHz, DMSO-$d_6$) δ: 2.23−2.40 (m, 2H, CH$_2$CO), 2.60−2.75 (m, 2H, CH$_2$CO), 3.17−3.38 (m, 4H, 2x NCH$_2$), 6.50–7.18 (m, 8H, H$_{Ar}$); 7.69–8.07 (m, 4H, H$_{Ar}$, N=CH); 8.95 (s, 1H, OH), 11.25, 11.27, 11.34 (3s, 2H, 2x NH); $^{13}$C NMR (101 MHz, DMSO-$d_6$) δ: 30.2, 30.3 ($\underline{C}$H$_2$CO), 47.9, 48.1, 48.6, 48.9 (NCH$_2$), 112.0, 112.1, 112.9, 113.2, 115.6, 115.7, 119.2, 119.3, 122.0, 122.5, 123.2, 124.2, 124.5, 124.9, 133.1, 135.9, 136.0, 136.2, 136.6, 136.9, 144.7, 144.8, 144.9, 149.2, 149.4, 152.2, 152.5, 152.8, 167.5, 167.6 (C$_{Ar}$), 173.3 (2x C=O). IR (KBr): ν$_{max}$ (cm$^{-1}$) = 3227 (OH), 3098 (2x NH), 1671 (2x C=O). Anal. Calcd. for C$_{22}$H$_{23}$N$_5$O$_5$, %: C 60.40; H 5.30; N 16.01. Found: C 60.27; H 5.17; N 15.87. HRMS m/z calculated for C$_{22}$H$_{23}$N$_5$O$_5$ [M+H]+ : 438.1699; found: 438.1770.

*3,3'-((2-Hydroxyphenyl)azanediyl)bis(N'-(thiophen-2-ylmethylene)propanehydrazide)* (*16*): White powder, yield 0.49 g (70%), m.p. 212−214 °C (from dioxane). $^1$H NMR (400 MHz, DMSO-$d_6$) δ: 2.31 and 2.68 (2t, $J$ = 6.9 Hz, 4H, CH$_2$CO), 3.15−3.34 (m, 4H, 2x NCH$_2$), 6.67−7.63 (m, 10H, H$_{Ar}$, H$_{Het}$), 8.12, 8.14, 8.33 (3s, 2H 2x N=CH), 8.95 (t, $J$ = 10.9 Hz, 1H, OH), 11.27, 11.29, 11.35 (3s, 2H, 2x NH); $^{13}$C NMR (101 MHz, DMSO-$d_6$) δ: 30.1, 30.4, 32.4, 32.5 ($\underline{C}$H$_2$CO), 48.1, 48.5, 48.6, 48.9 (NCH$_2$), 115.6, 115.7, 119.1, 119.2, 119.3, 122.3, 122.7, 123.1, 124.3, 124.5, 124.8, 127.8, 128.1, 128.2, 128.7, 129.9, 130.1, 130.6, 130.7, 136.2, 136.5, 136.8, 138.0, 138.1, 138.9, 139.0, 139.1, 141.2, 141.3, 152.3, 152.5,

152.7, 167.4, (C$_{Ar}$), 173.1 (2x C=O). IR (KBr): ν$_{max}$ (cm$^{-1}$) = 3203 (OH), 3011 (2x NH), 1672 (2x C=O). Anal. Calcd. for C$_{22}$H$_{23}$N$_5$O$_3$S$_2$, %: C 56.27; H 4.94; N 14.91. Found: C 52.05; H 4.74; N 14.76. HRMS m/z calculated for C$_{22}$H$_{23}$N$_5$O$_3$S$_2$ [M+H]+ : 470.1242; found: 470.1314.

*3,3'-((2-Hydroxyphenyl)azanediyl)bis(N'-((5-nitrothiophen-2-yl)methylene)propanehydrazide) (17)*: Yellow powder, yield 0.6 g (72%), m.p. 221−223 °C (from methanol). $^1$H NMR (400 MHz, DMSO-$d_6$) δ: 2.35 and 2.71 (2t, *J* = 7.2 Hz, 4H, CH$_2$CO), 3.16−3.47 (m, 4H, 2x NCH$_2$), 6.72−6.87 (m, 2H, H$_{Ar}$); 6.88−6.98 (m, 1H, H$_{Ar}$); 7.07−7.15 (m, 1H, H$_{Ar}$); 7.33−7.50 (m, 2H, H$_{Ar}$); 7.94−8.39 (m, 4H, H$_{Ar}$, N=CH); 8.81, 8.86, 9.02 (3s, 1H, OH), 11.68, 11.70, 11.74, 11.76 (4s, 2H, 2x NH); $^{13}$C NMR (101 MHz, DMSO-$d_6$) δ: 30.0, 30.3, 32.5, 32.7 (CH$_2$CO), 48.1, 48.6 (NCH$_2$), 115.6, 115.7, 119.1, 119.3, 122.4, 122.9, 124.7, 128.7, 128.8, 129.3, 130.4, 136.0, 136.2, 136.5, 139.5, 146.8, 146.9, 150.3, 150.6, 152.3, 152.5, 168.1 (C$_{Ar}$), 173.6, 173.7 (2x C=O). IR (KBr): ν$_{max}$ (cm$^{-1}$) = 3198 (OH), 3100 (2x NH), 1672 (2x C=O). Anal. Calcd. for C$_{22}$H$_{21}$N$_7$O$_7$S$_2$, %: C 47.22; H 3.78; N 17.52. Found: C 47.05; H 3.56; N 17.34.

*3,3'-((2-Hydroxyphenyl)azanediyl)bis(N'-(5-nitrofuran-2-ylmethylene)propanehydrazide) (18)*: Yellow powder, yield 0.47 g (59%), m.p. 188−190 °C (from dioxane). $^1$H NMR (400 MHz, DMSO-$d_6$) δ: 2.24−2.42 (m, 2H, CH$_2$CO), 2.74 (t, *J* = 6.7 Hz, 2H, CH$_2$CO), 3.20−3.41 (m, 4H, 2x NCH$_2$), 6.42−7.21 (m, 6H, H$_{Ar}$), 7.58−8.12 (m, 4H, H$_{Ar}$ 2x N=CH), 8.84, 8.89, 9.01 (3s, 1H, OH), 11.68, 11.72, 11.76, 11.79 (4s, 2H, 2x NH); $^{13}$C NMR (101 MHz, DMSO-$d_6$) δ: 30.1, 30.3, 32.5 (CH$_2$CO), 47.9, 48.4, 48.8 (NCH$_2$), 114.2, 114.4, 114.6, 114.7, 114.9, 115.5, 115.6, 115.8, 122.2, 122.7, 123.2, 124.3, 124.6, 124.9, 130.9, 131.0, 133.8, 133.9, 136.1, 136.4, 136.8, 151.6, 151.7, 151.8, 151.9, 152.2, 152.5, 168.1, 168.2 (C$_{Ar}$), 173.8, 173.9 (2x C=O). IR (KBr): ν$_{max}$ (cm$^{-1}$) = 3212 (OH), 3153 (2x NH), 1676 (2x C=O). Anal. Calcd. for C$_{22}$H$_{21}$N$_7$O$_9$, %: C 50.10; H 4.01; N 18.59. Found: C 49.94; H 3.87; N 18.35. HRMS m/z calculated for C$_{22}$H$_{21}$N$_7$O$_9$ [M+H]+ : 528.1400; found: 528.1476.

*3,3'-((2-Hydroxyphenyl)azanediyl)bis(N'-(thiophen-3-ylmethylene)propanehydrazide) (19)*: White powder, yield 0.62 g (89%), m.p. 209−211 °C (from 2-propanol). $^1$H NMR (400 MHz, DMSO-$d_6$) δ: 2.31 and 2.72 (2t, *J* = 7.2 Hz, 4H, CH$_2$CO), 3.18−3.32 (m, 4H, NCH$_2$), 6.72−8.21 (m, 12H, H$_{Ar}$, H$_{Het,}$ 2x CH), 8.90, 8.95, 8.98 (3s, 1H, OH), 11.19, 11.28 (2s, 2H, 2x NH); $^{13}$C NMR (101 MHz, DMSO-$d_6$) δ: 30.2, 30.4, 32.4, 32.5 (CH$_2$CO), 48.3, 48.5, 48.7, 48.9 (NCH$_2$), 115.5, 115.6, 115.7, 119.1, 119.2, 119.3, 122.4, 122.7, 123.1, 124.4, 124.5, 124.6, 124.7, 124.8, 127.3, 127.4, 127.5, 127.9, 136.3, 136.5, 136.9, 137.4, 137.5, 138.6, 138.7, 141.7, 141.8, 152.4, 152.6, 152.7, 167.4, 167.5 (C$_{Ar}$), 173.3 (2x C=O). IR (KBr): ν$_{max}$ (cm$^{-1}$) = 3287 (OH), 3088 (2x NH), 1670 (2x C=O). Anal. Calcd. for C$_{22}$H$_{23}$N$_5$O$_3$S$_2$, %: C 56.27; H 4.94; N 14.91. Found: C 56.05; H 4.71; N 14.84. HRMS m/z calculated for C$_{22}$H$_{23}$N$_5$O$_3$S$_2$ [M+H]+ : 470.1242; found: 470.1313.

*3,3'-((2-Hydroxyphenyl)azanediyl)bis(N'-(propan-2-ylidene)propanehydrazide) (20)*: White powder, yield 0.43 g (80%), m.p. 161−163 °C (from 2-propanol). $^1$H NMR (400 MHz, DMSO-$d_6$) δ: 1.81, 1.82, 1.86, 1.90 (4s, 12H, 4x CH$_3$); 2.32 (q, *J* = 6.9 Hz, 2H, CH$_2$CO), 2.60 (t, *J* = 6.9 Hz, 2H, COCH$_2$), 3.20 (q, *J* = 6.9 Hz, 4H, NCH$_2$), 6.70−6.81 (m, 2H, H$_{Ar}$), 6.85−6.95 (m, 1H, H$_{Ar}$), 7.03−7.11 (m, 1H, H$_{Ar}$), 8.93, 8.96, 9.01 (3s, 1H, OH), 10.02, 10.04 (2s, 2H, 2x NH); $^{13}$C NMR (101 MHz, DMSO-$d_6$) δ: 17.0, 17.5, 24.9, 25.1 (4x CH$_3$), 30.5, 30.6, 31.9, 32.0 (CH$_2$CO), 47.9, 48.2, 48.8, 48.9 (NCH$_2$), 115.6, 118.9, 119.1, 121.8, 122.3, 122.8, 124.1, 124.4, 124.6, 136.4, 136.6, 137.1, 150.4, 152.3, 152.6, 154.8, 154.9, 167.5 (C$_{Ar}$), 173.5 (2x C=O). IR (KBr): ν$_{max}$ (cm$^{-1}$) = 3271 (OH), 3198 (2x NH), 1660 (2x C=O). Anal. Calcd. for C$_{18}$H$_{27}$N$_5$O$_3$, %: C 59.81; H 7.53; N 19.38. Found: C 59.63; H 7.36; N 19.17. HRMS m/z calculated for C$_{18}$H$_{27}$N$_5$O$_3$ [M+H]+ : 362.2113; found: 362.2186.

*3,3'-((2-Hydroxyphenyl)azanediyl)bis(N'-(butan-2-ylidene)propanehydrazide) (21)*: Light brown powder, yield 0.49 g (84%), m.p. 81−83 °C (from 2-propanol). $^1$H NMR (400 MHz, DMSO-$d_6$) δ: 0.82−1.11 (m, 6H, CH$_2$CH3); 1.79, 1.81, 1.84, 1.88 (4s, 6H, 2x CH$_3$); 2.09−2.29 (m, 4H, CH2CH$_3$); 2.32 (t, *J* = 7.1 Hz, 2H, CH$_2$CO), 2.63 (t, *J* = 7.1 Hz, 2H, COCH$_2$), 3.21 (q, *J* = 6.9 Hz, 4H, NCH$_2$), 6.67−6.82 (m, 2H, H$_{Ar}$), 6.85−6.95 (m, 1H, H$_{Ar}$), 7.03−7.11 (m, 1H, H$_{Ar}$), 8.95 (1s, 1H, OH), 10.01, 10.09 (2s, 2H, 2x NH); $^{13}$C NMR (101 MHz, DMSO-$d_6$) δ: 9.7, 9.8, 10.5, 10.8, 15.7, 15.9 (4x CH$_3$), 22.1, 22.9, 23.3 (CH$_2$CH$_3$), 30.5, 31.4, 31.7, 31.9, 32.1 (CH$_2$CO), 48.1, 48.5, 48.8, 48.9 (NCH$_2$), 115.4, 115.5, 119.1, 122.1, 122.4, 122.7, 124.2, 124.5, 124.6, 136.6, 137.0, 152.4, 152.5, 153.7, 153.8, 158.3, 167.6 (C$_{Ar}$), 173.72 (2x C=O). IR (KBr): ν$_{max}$ (cm$^{-1}$) =

3240 (OH), 3178 (2x NH), 1671 (2x C=O). Anal. Calcd. for $C_{20}H_{31}N_5O_3$, %: C 61.67; H 8.02; N 17.98. Found: C 61.45; H 7.88; N 17.87. HRMS m/z calculated for $C_{20}H_{31}N_5O_3$ [M+H]+ : 390.2426; found: 390.2500.

*3,3'-((2-Hydroxyphenyl)azanediyl)bis(N'-(1-phenylethylidene)propanehydrazide) (22)*: White powder, yield 0.61 g (84%), m.p. 187−189 °C (from 2-propanol). $^1$H NMR (400 MHz, DMSO-$d_6$) δ: 2.09–2.27 (m, 6H, $CH_3$); 2.40–2.55 (overlaps with DMSO, 2H, $CH_2CO$), 2.80 (t, *J*=7.1 Hz, 2H, $COCH_2$), 3.22–3.42 (m, 4H, $NCH_2$), 6.68–6.99 (m, 3H, $H_{Ar}$), 7.07–7.18 (m, 1H, $H_{Ar}$), 7.27–7.47 (m, 6H, $H_{Ar}$), 7.52–7.85 (m, 4H, $H_{Ar}$); 8.93, 8.99 (2s, 1H, OH), 10.44, 10.45, 10.48, 10.50 (4s, 2H, 2x NH); $^{13}$C NMR (101 MHz, DMSO-$d_6$) δ: 13.5, 14.0 (2x $CH_3$), 30.6, 30.8, 32.2, 32.4 ($CH_2CO$), 48.5, 48.7, 48.9 ($NCH_2$), 115.5, 115.6, 119.1, 119.2, 122.5, 122.8, 124.4, 125.9, 126.3, 128.2, 128.3, 128.8, 128.9, 129.1, 136.4, 136.9, 138.1, 138.3, 147.2, 147.3, 150.9, 152.6, 168.1, 168.2 ($C_{Ar}$), 174.3 (2x C=O). IR (KBr): $v_{max}$ ($cm^{-1}$) = 3242 (OH), 3158 (2x NH), 1677 (2x C=O). Anal. Calcd. for $C_{28}H_{31}N_5O_3$, %: C 69.26; H 6.44; N 14.42. Found: C 69.03; H 6.21; N 14.24.

*3,3'-((2-Hydroxyphenyl)azanediyl)bis(N-(2,5-dimethyl-1H-pyrrol-1-yl)propanamide) (23)*: To a solution of dihydrazide **5** (0.5 g, 1.8 mmol) in 2-propanol (25 mL), hexane-2,5-dione (0.82 g, 7.2 mmol) and a catalytic amount of acetic acid (0.1 mL) were added, and the mixture was heated under reflux for 6 h, then cooled down, and was diluted with water (25 mL); the formed precipitate was filtered off, washed with water, and recrystallized from a mixture of 2-propanol and water. White powder, yield 0.53 g (67%), m.p. 200−202 °C (from 2-propanol). $^1$H NMR (400 MHz, DMSO-$d_6$) δ: 1.95 (s, 12H, 4x $CH_3$); 2.41 (t, *J*=7.2 Hz, 4H, 2x $COCH_2$), 3.34 (t, *J*=7.2 Hz, 4H, 2x $NCH_2$), 5.61 (s, 4H, 4x $CH_{pyr}$); 6.74–6.86 (m, 2H, $H_{Ar}$), 6.90–6.98 (m, 1H, $H_{Ar}$), 7.09–7.16 (m, 1H, $H_{Ar}$), 8.88 (1s, 1H, OH), 10.60 (1s, 2H, 2x NH); $^{13}$C NMR (101 MHz, DMSO-$d_6$) δ: 10.9 (2x $CH_3$), 31.4 ($CH_2CO$), 47.8 ($NCH_2$), 102.8, 115.7, 119.3, 123.3, 124.8, 126.7, 136.1, 152.6 ($C_{Ar}$), 170.6 (C=O). IR (KBr): $v_{max}$ ($cm^{-1}$) = 3237 (OH), 3018 (2x NH), 1672 (2x C=O). Anal. Calcd. for $C_{24}H_{31}N_5O_3$, %: C 65.88; H 7.14; N 16.01. Found: C 65.63; H 6.95; N 15.87. HRMS m/z calculated for $C_{24}H_{31}N_5O_3$ [M+H]+ : 438.2426; found: 438.2498.

*5-(3-(3,5-Dimethyl-1H-pyrazol-1-yl)-3-oxopropyl)-4,5-dihydrobenzo[b][1,4]oxazepin-2(3H)-one (24)*: To a solution of dihydrazide **5** (0.5 g, 1.8 mmol) in 2-propanol (28 mL), pentane-2,4-dione (0.9 g, 9.0 mmol) and a catalytic amount of hydrochloric acid (0.05 mL) were added, and the mixture was heated under reflux for 5 h, then cooled down. The solvent was removed under reduced pressure, the residue was poured with water (30 mL), and the formed precipitate was filtered off, washed with water and diethyl ether, and was recrystallized from a mixture of 2-propanol and water. White powder, yield 0.35 g (62%), m.p. 133−135 °C (from 2-propanol). $^1$H NMR (400 MHz, DMSO-$d_6$) δ: 2.12 and 2.44 (2s, 6H, 2x $CH_3$); 2.56 and 3.22 (2t, *J*=6.9 Hz, 4H, 2x $COCH_2$), 3.38 and 3.44 (2t, *J*=6.9 Hz, 4H, 2x $NCH_2$), 6.14 (s, 1H, 4x $CH_{Het}$); 7.00–7.33 (m, 4H, $H_{Ar}$); $^{13}$C NMR (101 MHz, DMSO-$d_6$) δ: 13.4, 14.0 (2x $CH_3$), 31.6, 33.5 ($CH_2CO$), 46.7, 54.5 ($NCH_2$), 111.1, 119.5, 121.0, 123.9, 126.2, 138.2, 143.1, 146.9, 151.3, 169.9 ($C_{Ar}$), 171.6 (C=O). IR (KBr): $v_{max}$ ($cm^{-1}$) = 1748, 1729 (2x C=O). Anal. Calcd. for $C_{17}H_{19}N_3O_3$, %: C 65.16; H 6.11; N 13.41. Found: C 64.98; H 5.96; N 13.27. HRMS m/z calculated for $C_{17}H_{19}N_3O_3$ [M+Na]+ : 336.1426; found: 336.1321.

*3,3'-((2-Hydroxyphenyl)azanediyl)bis(N'-((Z)-2-oxoindolin-3-ylidene)propanehydrazide) (25)*: To a solution of hydrazide **5** (0.3, 1.06 mmol) in methanol (10 mL), isatin (0.38, 2.55 mmol) and glacial acetic acid (1 drops) were added. The reaction mixture was heated under reflux for 4 h. Precipitate was filtered off, washed with methanol, and recrystallized from 2-propanol/$H_2O$ mixture. Yellow powder, yield 0.48 g (83%), m.p. 156−158 °C (2-propanol/H2O mixture). $^1$H NMR (400 MHz, DMSO-$d_6$) δ: 2.57–2.99 (m, 4H, 2x $COCH_2$), 3.39 (t, *J*=7.1 Hz, 4H, 2x $NCH_2$), 6.99–7.40 (m, 10H, $H_{Ar}$), 7.82–8.11 (m, 2H, $H_{Ar}$), 8.94 (s, 1H, OH), 10.75 and 11.09 (2s, 4H, 4x NH); $^{13}$C NMR (101 MHz, DMSO-$d_6$) δ: 31.6 ($CH_2CO$), 48.2 ($NCH_2$), 110.5, 115.3, 115.7, 119.3, 121.6, 124.4, 126.1, 126.2, 132.4, 143.6, 152.2, 164.6 ($C_{Ar}$), 174.8, 184.9 (2x C=O). IR (KBr): $v_{max}$ ($cm^{-1}$) = 3220 (OH), 3202 (4x NH), 1682, 1618 (2x C=O). Anal. Calcd. for $C_{28}H_{25}N_7O_5$, %: C 62.33; H 4.67; N 18.17. Found: C 62.17; H 4.42; N 17.89. HRMS m/z calculated for $C_{28}H_{25}N_7O_5$ [M+Na]+ : 540.1917; found: 540.1995.

*5,5'-(((2-Hydroxyphenyl)azanediyl)bis(ethane-2,1-diyl))bis(1,3,4-oxadiazole-2(3H)-thione) (26)*: A mixture of dihydrazyde **5** (0.7 g, 2.5 mmol), potassium hydroxide (2.19 g, 39 mmol), carbon disulfide (3.62 g, 47.5 mmol), and 60 mL methanol was refluxed for 24 h, and then the volatile fractions were separated under reduced pressure. The obtained residue was dissolved in water (20 mL), and the solution was acidified with acetic acid to pH 6. The formed solid was filtered off,

washed with water, and recrystallized from a mixture of 2-propanol and water. White powder, yield 0.65 g (71%), m.p. 176−178 °C (from 2-propanol). $^1$H NMR (400 MHz, DMSO-$d_6$) δ: 2.79 (t, $J$ = 7.1 Hz, 4H, 2x COCH$_2$), 3.40 (t, $J$ = 7.1 Hz, 4H, 2x NCH$_2$), 6.73 (t, $J$ = 7.5 Hz, 1H, H$_{Ar}$), 6.79 (d, $J$ = 7.8 Hz, 1H, H$_{Ar}$), 6.90 (t, $J$ = 7.5 Hz, 1H, H$_{Ar}$), 6.97 (d, $J$ = 7.8 Hz, 1H, H$_{Ar}$), 8.97 (s, 1H, OH), 14.24 (br s, 2H, 2x NH); $^{13}$C NMR (101 MHz, DMSO-$d_6$) δ: 23.9 (CH$_2$CO), 48.5 (NCH$_2$), 115.9, 119.2, 123.4, 124.7, 135.2, 152.3, 162.8 (C$_{Ar}$), 177.6 (2x C=S). IR (KBr): ν$_{max}$ (cm$^{-1}$) = 3193 (OH), 2966 (2x NH). Anal. Calcd. for C$_{14}$H$_{15}$N$_5$O$_3$S$_2$, %: C 46.02; H 4.14; N 19.17. Found: C 45.87; H 3.89; N 18.96. HRMS m/z calculated for C$_{14}$H$_{15}$N$_5$O$_3$S [M+Na]+: 388.0616; found: 388.0508.

**X-ray crystallography.** Diffraction data of **24** were collected at 160 K on a Rigaku, XtaLAB Synergy, Dualflex, HyPix diffractometer using monochromatic Cu-Kα radiation (λ = 1.54184 Å). The crystal structure was solved using the direct method and refined with the ShelXLrefinement package using Least Squares minimization. All nonhydrogen atoms were refined in anisotropic approximation. The hydrogen atoms involved in the formation of H-bonds were refined isotopically; all other H-atoms were refined by riding model with $U$iso(H) = 1.2$U$eq(C). Crystal data: monoclinic, $a$ = 8.42179(6), $b$ = 22.5776(2), $c$ = 8.60767(7) Å, β = 109.3386(8)°; $V$ = 1544.35(2) Å$^3$, Z = 4, μ = 0.770 mm$^{-1}$, Dcalc = 1.348 g·cm$^{-3}$; space group is $P2_1/n$. The final $R_1$ was 0.0344 ($I$ > 2σ($I$)) and $wR_2$ was 0.0909 (all data). For further details, see crystallographic data for this compound deposited at the Cambridge Crystallographic Data Centre. Deposition Number (https://www.ccdc.cam.ac.uk/services/structures) CCDC 2420814 contains the supplementary crystallographic data for this paper. These data are provided free of charge by the joint Cambridge Crystallographic Data Centre and Fachinformationszentrum Karlsruhe Access Structures service.

**Microbial strains and culture conditions.** The multidrug-resistant *Stahphylococcus aureus* strain TCH 1516 [USA 300-HOU-MR] and pan-susceptible *S. aureus* ATCC 25923 was obtained from the American Type Culture Collection (ATCC). *Acinetobacter baumannii* PKC-1027 and *Enterobacter cloacae*PKC-0122 are laboratory strains of clinical origin obtained from the Institute of Infectious Diseases and Pathogenic Microbiology collection (Prienai, Lithuania). *Enterococcus faecalis* AR-0781, *Klebsiella pneumoniae* AR-0153 and *Pseudomonas aeruginosa* AR-0054 were obtained from ARisolate bank at CDC Atlanta. *Candida albicans* AR-0761 and *Candida auris* AR-0381 were obtained from ARisolate bank at CDC Atlanta. *Aspergillus fumigatus* CEA10, *Cuninghamella bertholiatiae* NIH-182, and *Rhizopus delamar* 1221 were kindly provided by Dr. Vidmantas Petraitis (Weill Cornell Medicine of Cornell University). *Candida krusei* ATCC 32196 were obtained from ATCC. Prior to the experiments all microbial strains were stored in commercial cryopreservation systems at a temperature of −80 °C. The strains were cultivated on Columbia sheep blood agar for bacterial strains (Becton Dickinson, Franklin Lakes, NJ, USA). Fungal strains were cultured on Sabouraud Dextrose agar (Becton Dickinson, Franklin Lakes, NJ, USA).

**Minimal inhibitory concentration determination.** The antimicrobial activity of synthesized compounds or control antibiotics was assessed using the broth microdilution method, following the guidelines outlined by the Clinical Laboratory Standards Institute (CLSI), with modifications [29–31]. The test compounds were dissolved in dimethylsulfoxide (DMSO) to obtain a final concentration of 25–30 mg/mL. Vancomycin hydrochloride and gentamycin sulphate were dissolved in sterile deionized water, while meropenem and cefazolin were dissolved in DMSO (MedChemExpress, Deer Park, United Stated). Dilution series were prepared in deep 96-well microplates (Nunc, Thermo Scientiffic, Waltham, United States) to achieve a two-fold concentration range of 0.5, 1, 2, 4, 8, 16, 32, 64 and 128 µg/mL, utilizing cation-adjusted Mueller–Hinton broth (CAMHB) (Thermo Scientiffic, Waltham, United States) as the growth medium. For *Candida* or filamentous mold screening, dilutions of test compounds were performed in RPMIMOPS broth (Criterion, Hardy Diagnostics, West McCoy Lane Santa Maria). Amphotericin B, fluconazole (Sigma, St. Louis, United States), posaconazole, and voriconazole (MedChemExpress, Deer Park, United Stated) were dissolved in DMSO and further serially diluted using RPMI/MOPS broth. The microplates containing the dilution series were then inoculated with fresh cultures of each tested organism to reach a final concentration of 5 × 10$^4$ CFU (colony-forming units) of the test organism in media containing 1% DMSO and 1 × compound or control antimicrobial concentration, with a volume of 200 µL per well. Wells that were

inoculated with media containing 1% DMSO served as positive controls. Subsequently, the microplates were incubated at 35±1 °C for 18±2h. Following the incubation period, the plates were examined using a manual microplate viewer (Sensititre Manual Viewbox, United States). The minimal inhibitory concentration (MIC) was defined as the lowest concentration (µg/mL) of the tested drug that completely inhibited the growth of the test organism. All experiments were conducted in duplicate with three technical replicates for each condition.

**Minimal bactericidal concentration determination.** Following the determination of the minimum inhibitory concentration (MIC) for the test compounds and control antimicrobial agents, 10 µL aliquots were removed from each well of the 96-well microplates and transferred onto Columbia sheep blood agar plates (Becton Dickinson, Franklin Lakes, NJ, USA). The inoculated plates were allowed to dry for 10 minutes under laminar airflow to prevent pooling of the samples. Subsequently, the plates were inverted and incubated at 37 °C for 18hours to allow for microbial growth.

The minimal bactericidal concentration (MBC) was defined as the lowest concentration of the test compound or control antibiotic that resulted in no visible colonies on the agar plate, indicating complete bacterial killing. Each experiment was performed in triplicate.

**Time-kill assay.** Prior to the time-kill experiments, E. faecalis AR-0781 and S. aureus TCH-1516 were subcultured on Mueller-Hinton agar to obtain well-separated colonies. One to two colonies were picked and suspended in 5mL of cation-adjusted Mueller-Hinton broth (CAMHB) and cultured overnight at 37 °C. The overnight cultures were diluted 1:100 in 5mL of fresh, pre-warmed CAMHB and incubated at 37 °C with shaking at 200rpm until the $OD_{600}$ reached 0.5. The cultures were then diluted 1:50 in CAMHB containing 0.1% DMSO, which served as the compound-free control, or CAMHB supplemented with 0.1% DMSO and either compound **9** or compound **26** at sub-MIC, MIC and 1X MIC concentrations. The cultures were incubated at 37°C with shaking at 200rpm, and at 2, 4, 6, and 24-hour time points, a 100 µL aliquot was taken, serially diluted, and plated on sheep blood agar. The plates were incubated overnight at 37°C, and the resulting colonies were counted.

## Results

### Synthesis of *N*-substituted *β*-amino acid derivatives

In the first stage of this work, by using a well-known methodology described in the publication [32], the initial compound **5** were prepared. According to the methodology, the reaction of 2-aminophenol (**1**) with acrylic acid in water at reflux afforded intermediates 3,3′-((2-hydroxyphenyl)azanediyl)di(propanoic)acid (**2**) (Scheme 1).

**Scheme 1. Synthesis of compounds 2–6.**

The attempt to synthetize the diamide **3** led to unexpected product. However, the diacid **2** cyclizes under these reaction conditions to the oxadiazepine moiety bearing derivative **3**. Compound **3** was prepared by direct coupling of acids **2** with the sulphanilamide using HBTU as the coupling reagent and triethylamine as the base. The reaction were performed in dimethylformamide at room temperature. The product **3** was isolated by the dilution of the reaction mixture with water and were characterised using NMR, HRMS, IR spectroscopy and elemental analysis. Comparison of the $^1$H NMR spectra of product **3** with the compound **2** has revealed, that proton singlet of hydroxy group at 8.77 ppm in the spectra of diacid **2** have been dissapered with the formation of the oxazepine fragment. The structure of the obtained compound **3** is also confirmed by the results of the HRMS m/z calculated for $C_{18}H_{19}N_3O_5S$ [M+H]$^+$: 390.1045; found: 390.1119.

In continuation of our interest in the chemistry of *N*-substituted *β*-amino acids, dimethyl ester **4** was synthesized through esterification of 3,3′-((2-hydroxyphenyl)azanediyl)di(propanoic)acid (**2**) with an excess of methanol in the presence of a catalytic amount of sulfuric acid. Dihydrazide **5** was obtained through hydrazinolysis of dimethyl ester **4** in propan-2-ol under reflux.

Condensation of dihydrazide **5** with aromatic aldehydes and ketones gave the corresponding hydrazones **7–22**. The structures of hydrazones **7–22** have been established mainly on the basis of $^1$H and $^{13}$C NMR spectra (Scheme 2). The restricted rotation around the CONH led to the formation in an isomeric mixture of hydrazones where Z isomer

**7** Ar=C$_6$H$_5$; **8** Ar=2,4-F-C$_6$H$_3$; **9** Ar=4-NO$_2$-C$_6$H$_4$; **10** Ar=4-Cl-C$_6$H$_4$; **11** Ar=4-(CH$_3$)$_2$N-C$_6$H$_4$; **12** Ar=4-HO-C$_6$H$_4$; **13** Ar=3,4,5-(OCH$_3$)$_3$-C$_6$H$_2$; **14** Ar=1-naftil; **15** Het= 2-furyl; **16** Het= 2-thienyl; **17** Het= 5-nitro-2-thienyl; **18** Het= 5-nitro-2-furyl; **19** Het= 3-thienyl; **20** R=CH$_3$; **21** R=C$_2$H$_5$; **22** R=C$_6$H$_5$;

**Scheme 2. Synthesis of compounds 7–22.**

predominates. The obtained hydrazones **7−22** show double sets of resonances for the N=CH and CONH fragment protons with the intensity ratio of 0.3:0.7 (1H NMR). No formation of geometrical isomers wasobserved.

In the next stage of this work, condensation reactions of dihydrazide **5** with various dicarbonyl compounds and carbon disulfite were performed. The reaction of dihydrazide **5** with hexane-2,5-dione, isatin and carbon disulfide resulted in the formation of compounds **23, 25** and **26** of the expected structure, each containing two identical heterocyclic fragments, while in the reaction with acetylacetone, cyclization also took place with the participation of the hydroxy group in the *o*-position, forming the corresponding 5-(3-(3,5-dimethyl-1*H*-pyrazol-1-yl)-3-oxopropyl)-4,5-dihydrobenzo[*b*][*1,4*]oxazepin-2(3*H*)-one **(24)** (Scheme 3).

Comparison of the ¹H NMR spectra of product **24** with the compound **5** has revealed, that proton singlet of hydroxy group at 9.01 ppm in the spectra of dihydrazide **5** have been dissapered with the formation of the oxazepine fragment. The structure of the obtained compound **24** is also confirmed by the results of the HRMS – m/z calculated for $C_{17}H_{19}N_3O_3$ [M+Na]⁺: 336.1426; found: 336.1321 (S1–S50 Figs in S1 File).

### X-ray crystallographic study

To confirm the structure of synthesized compound, we performed an X-ray diffraction analysis on compound **24** (Fig 2 and S2–S8 Tables in S1 File). The molecular structure is characterized by fully staggered conformation of -CH$_2$−CH$_2$- fragment; the torsion angle of N5−C11−C12−C13 is equal −161.9(1)°. The conformation of the seven-membered cycle is close to the boat; the deviations of the C3, C5a and C9a atoms from the least-square plane of O1, C2, C4, N5 are 0.663(2), 1.008(1) and 0.901(2) Å, respectively. Carbon atoms C4 and C11 connected to nitrogen N5 have increased electronegativity; due to this, weak intermolecular hydrogen bonds of CH···O type are formed in the crystal structure. The parameters of these bonds are follows: C4···O1 = 3.375(2) Å, H···O1 = 2.51(2) Å, C4−H···O1 = 145(1)°; C11···O10 = 3.699(2) Å, H···O10 = 2.74(2) Å, C11−H···O10 = 166(1)°. By means of these bonds, molecular chains are formed in the crystal structure along the crystallographic direction [1 0 1].

### 2-Hydroxyphenyl propanoic acid derivatives shows Gram-positive bacteria-directed activity

After successfully synthesizing and characterizing a series of 2-hydroxyphenyl propanoic acid derivatives (compounds **2–26**), we evaluated their *in vitro* antimicrobial activity by determining minimal inhibitory concentration (MIC) as well as minimal bactericidal concentration (MBC). The compounds were screened using a laboratory strain collection of ESKAPE

**Scheme 3. Synthesis of compounds 23–26.**

pathogens, such as *Staphylococcus aureus, Klebsiella pneumoniae, Acinetobacter baumannii, Pseudomonas aeruginosa,* and *Enterobacter cloacae.* Due to rising antimicrobial resistance among *Enterococcus faecalis*, we selected a vancomycin resistant isolate and included it in our screening. These pathogens were selected due to their clinical significance and the presence of genetically defined and emerging antimicrobial resistance mechanisms, while the control antibiotics (vancomycin, gentamycin, meropenem, and cefazolin) were selected to represent the antibacterial agents used in the clinical setting to treat infections caused by Gram-negative and Gram-positive pathogens.

Starting compound **2** demonstrated no antimicrobial activity against tested bacterial and fungal strains (Table 1 and S1 Table in S1 File). Oxadiazepine derivative **3** demonstrated moderate antimicrobial activity against *Enterococcus faecalis* AR-0781 and *Staphylococcus aureus* TCH-1516, with a minimum inhibitory concentration (MIC) of 64 µg/mL. Dimethyl ester **4** showed no antimicrobial activity against any of the tested bacterial and fungal strains (Table 1 and S1 Table in S1 File). The dihydrazide **5** exhibited activity against *S. aureus* TCH-1516 (MIC 64 µg/mL) but not against *E. faecalis* AR-0781 or any tested Gram-negative pathogens. Benzimidazole **6** demonstrated moderate activity against *E. faecalis* AR-0781 (MIC 64 µg/mL) and weak activity against *S. aureus* TCH-1516 (MIC 128 µg/mL).

To characterize the *in vitro* structure-dependent effects of aromatic substitutions on antimicrobial activity, hydrazide **5** was used as a starting compound. The incorporation of an aryl substituent resulted in compound **7**, which showed no antimicrobial activity against all tested bacterial and fungal isolates (MIC > 128 µg/mL) (Table 1 and S1 Table in S1 File). The incorporation of a 2,4-difluorophenyl substituent (compound **8**) demonstrated activity against methicillin-resistant *S. aureus* TCH-1516 (MIC 32 µg/mL) (MRSA) but not vancomycin-resistant *E. faecalis* AR-0781 (MIC > 128 µg/mL) (VRE). Compound **8** also showed near-MIC range bactericidal activity against *S. aureus* TCH-1516 (MBC 64 µg/mL) (Table 1 and S2 Table in S1 File). Furthermore, compound **8** showed weak activity against the ESBL-producing *E. cloacae* PKC-0122 strain (MIC 128 µg/mL), but not other Gram-negative pathogens. In addition, the incorporation of a 4-nitrophenyl substituent (compound **9**) greatly enhanced the *in vitro* antimicrobial activity against *E. faecalis* AR-0781 (MIC 16 µg/mL) and *S. aureus* TCH-1516 (MIC 8 µg/mL) with MBC 16 µg/mL respectively, although it resulted in a loss of activity against *E. cloacae* PKC-0122 (MIC > 128 µg/mL) (S2 Table in S1 File). Furthermore, the addition of a 4-chlorophenyl (compound **10**) or 4-(dimethylamino)phenyl (compound **11**) substituent resulted in a loss of antimicrobial activity against all tested isolates (MIC > 128 µg/mL). The incorporation of a 4-hydroxyphenyl substituent (compound **12**) showed weak activity against *E. faecalis* AR-0781 (MIC 128 µg/mL) and favorable activity against *S. aureus* TCH-1516 (MIC 16 µg/mL) with MBC of 32 µg/mL (S2 Table in S1 File). The 3,4,5-trimethoxyphenyl derivative (compound **13**) showed no antimicrobial activity, while the

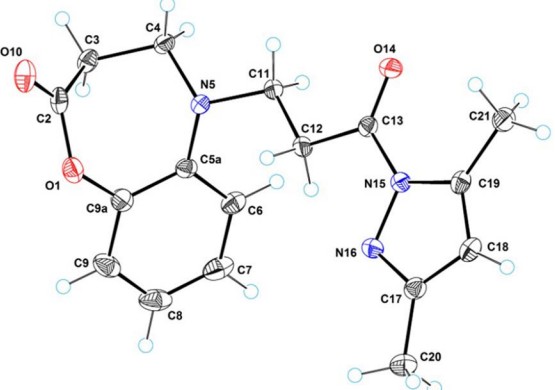

**Fig 2. The ORTEP diagram of analysed compound 24 showing the numbering scheme used in this study.**

addition of a 1-naphthyl substituent (compound **14**) resulted in favorable activity against *S. aureus* TCH-1516 (MIC 8 µg/mL) (Table 1).

To better understand the function of aromatic substitutions on antimicrobial activity, we further generated a series of compounds with heterocyclic substituents (compounds **15–19**). Interestingly, compounds bearing heterocyclic

**Table 1. The *in vitro* antimicrobial activity of *N*-substituted β-amino acid derivatives 2–26 against panel of multidrug-resistant bacterial strains.**

| Compound | Minimal inhibitory concentration (µg/mL) | | | | | |
|---|---|---|---|---|---|---|
| | *E. faecalis* AR-0781[A] | *S. aureus* TCH-1516[B] | *K. pneumo-niae* AR-0153[C] | *A. baumannii* PKC-1027[D] | *P. aeruginosa* AR-0054[E] | *E. cloacae* PKC-0122[F] |
| 2 | >128 | >128 | >128 | >128 | >128 | >128 |
| 3 | 64 | 64 | >128 | >128 | >128 | >128 |
| 4 | >128 | >128 | >128 | >128 | >128 | >128 |
| 5 | >128 | 64 | >128 | >128 | >128 | >128 |
| 6 | 64 | 128 | >128 | >128 | >128 | >128 |
| 7 | >128 | >128 | >128 | >128 | >128 | >128 |
| 8 | >128 | 32 | >128 | >128 | >128 | 128 |
| 9 | 16 | 8 | >128 | >128 | >128 | >128 |
| 10 | >128 | >128 | >128 | >128 | >128 | >128 |
| 11 | >128 | >128 | >128 | >128 | >128 | >128 |
| 12 | 128 | 16 | >128 | >128 | >128 | >128 |
| 13 | >128 | >128 | >128 | >128 | >128 | >128 |
| 14 | >128 | 8 | >128 | >128 | >128 | >128 |
| 15 | >128 | >128 | >128 | >128 | >128 | >128 |
| 16 | >128 | 64 | >128 | >128 | >128 | >128 |
| 17 | >128 | 8 | >128 | >128 | >128 | >128 |
| 18 | >128 | 16 | >128 | >128 | >128 | >128 |
| 19 | >128 | >128 | >128 | >128 | >128 | >128 |
| 20 | >128 | >128 | >128 | >128 | >128 | >128 |
| 21 | >128 | >128 | >128 | >128 | >128 | >128 |
| 22 | >128 | >128 | >128 | >128 | >128 | >128 |
| 23 | >128 | >128 | >128 | >128 | >128 | >128 |
| 24 | 128 | 32 | >128 | >128 | >128 | >128 |
| 25 | 64 | 32 | >128 | >128 | >128 | >128 |
| 26 | 8 | 4 | >128 | >128 | >128 | >128 |
| **Vancomycin** | 128 | 2 | N/A | N/A | N/A | N/A |
| **Gentamycin** | 16 | 32 | 64 | >128 | 128 | 64 |
| **Meropenem** | 2 | 2 | 64 | 64 | 64 | 2 |
| **Cefazolin** | 8 | 32 | >128 | >128 | >128 | 32 |

[A]Vancomycin-resistant *E. faecalis* harboring *tet(L), tet(M), VanA* resistance genes.

[B]Meticillin-Resistant *S. aureus* USA300 lineage harboring *mecA* and Panton-Valentine leucocidin *pvl*.

[C]Carbapenem-resistant *K. pneumoniae* harboring *aac(3)-IId, aadA2, armA, cmlA1, CTX-M-15, dfrA1, dfrA12, dfrA14, fosA, mph(E), msr(E), NDM-1, oqxA, OXA-1, OXA-232, OXA-9, strA, strB, sul1, sul2, TEM-1A* resistance genes.

[D]Colistin-resistant *A. baumannii* complex blood isolate.

[E]Carbapenem-resistant *P. aeruginosa* harboring *ant(2")-la, aph(3')-IIb, aph(6)-Id, bcr1, bcr1, bcr1, catB7, fosA, mexA, mexE, mexX, OXA-396, PDC-3, sul1, sul1, sul1, tet(A), tet(R), tet(R), tet(R), VIM-4* resistance genes. Extended-spectrum beta-lactamases (ESBL) producing blood isolate of *E. cloacea* harboring *CTX-M-15, aac(3)-IId, aadA2* resistance genes.

substitutions demonstrated activity only against the *S. aureus* TCH-1516 strain. Compound **15**, bearing a 2-furyl substituent, showed no antimicrobial activity (MIC > 128 µg/mL). The incorporation of a 2-thienyl substituent resulted in compound **16** with weak antimicrobial activity against the *S. aureus* TCH-1516 strain (MIC 64 µg/mL).

The incorporation of a 4-nitro group on the 2-thienyl (compound **17**) enhanced antimicrobial bactericidal activity against *S. aureus* TCH-1516 (MIC and MBC 8 µg/mL respectively), while the replacement of the 4-nitro-2-thienyl with a 5-nitro-2-furyl (compound **18**) reduced activity against *S. aureus* TCH-1516 (MIC 16 µg/mL) (Table 1 and S2 Table in S1 File). Surprisingly, the addition of a 3-thienyl substituent (compound **19**) diminished antimicrobial activity against *S. aureus* TCH-1516 (MIC > 128 µg/mL), demonstrating that the 2-thienyl position is crucial for *S. aureus* TCH-1516-directed activity. Finally, the methyl group adjacent substitutions resulted in compounds **20–22**, with no antimicrobial activity against the tested strains (Table 1).

To characterize the effect of heterocyclic substituents on the *in vitro* antimicrobial activity, we generated a series of compounds bearing known heterocyclic pharmacophores such as dimethylpyrole, isatin or thiosemicarbazide. Compound **23** bearing two identical dimethylpyrole substitutions demonstrated no antimicrobial activity against tested bacterial strains (MIC > 128 µg/mL). Oxadiazepine derivative **24** bearing dimethyl pyrazole substitution demonstrated weak activity against *E. faecalis* AR-0781(MIC 128 µg/mL) and favorable activity against *S. aureus* TCH-1516 (MIC 32 µg/mL). Compound **25** bearing isatin substituent showed activity against both *E. faecalis* AR-0781(MIC 64 µg/mL) and *S. aureus* TCH-1516 (MIC 32 µg/mL). Thiosemicarbazide derivative **26** showed potent antimicrobial activity against *E. faecalis* AR-0781(MIC 8 µg/mL) and *S. aureus* TCH-1516 (MIC 4 µg/mL) with MBC 16 and 8 µg/mL respectively (Table 1 and S2 Table in S1 File). Interestingly, compound **26** also demonstrated antifungal activity against *Candida albicans* AR-0671 strain (MIC 16 µg/mL) but not other *Candida* species or filamentous molds (S1 Table in S1 File).

These results demonstrates that *N*-substituted *β*-amino acid derivative **26** could be further explored as starting compound to generate a sub-library of compounds potentially targeting multidrug-resistant Gram-positive pathogens with emerging antimicrobial resistance mechanisms.

### Time-kill kinetics and antimicrobial activity of most promising compound 9 and compound 26 against drug-resistant *E. faecalis* AR-0781 and *S. aureus* TCH-1516

After identifying the most promising compounds (**9** and **26**), we aimed to characterize their time-kill kinetics using sub-MIC, MIC, and 1×MIC concentrations against vancomycin-resistant *E. faecalis* AR-0781 and methicillin-resistant (MRSA) *S. aureus* TCH-1516 strains (Fig 3).

Time-kill assays demonstrated that compound **9** exhibits dose-dependent bacteriostatic to bactericidal activity against *E. faecalis* AR-0781 (Fig 2). At 8 µg/mL and 16 µg/mL, compound **9** initially slowed bacterial growth but did not induce substantial killing. However, at the highest concentration tested (32 µg/mL), a significant reduction in bacterial burden was observed over time, suggesting a bactericidal effect at or above the MIC. In contrast, compound **26** displayed a more potent antimicrobial activity against *E. faecalis* AR-0781, with bactericidal activity observed at all tested concentrations (4, 8, and 16 µg/mL). The highest concentration (16 µg/mL) resulted in the most rapid and sustained bacterial killing. These results indicate that compound **26** exhibits more potent antimicrobial activity compared to compound **9** *against E. faecalis* (Fig 3).

In contrast, compound **9** displayed a weaker antimicrobial effect against *S. aureus* TCH-1516, compared to its activity against *E. faecalis* AR-0781. At the lowest concentration (4 µg/mL), minimal inhibition was observed, while at 8 and 16 µg/mL, bacterial burden was reduced but not eliminated, indicating that the compound is primarily bacteriostatic at these concentrations. Conversely, compound **26** demonstrated greater activity against *S. aureus* TCH-1516, with a clear dose-dependent reduction in bacterial burden. The highest tested concentration (8 µg/mL) achieved significant bacterial killing, suggesting that compound **26** shows more potent, Gram-positive bacteria directed activity than compound **9 (**Fig 3).

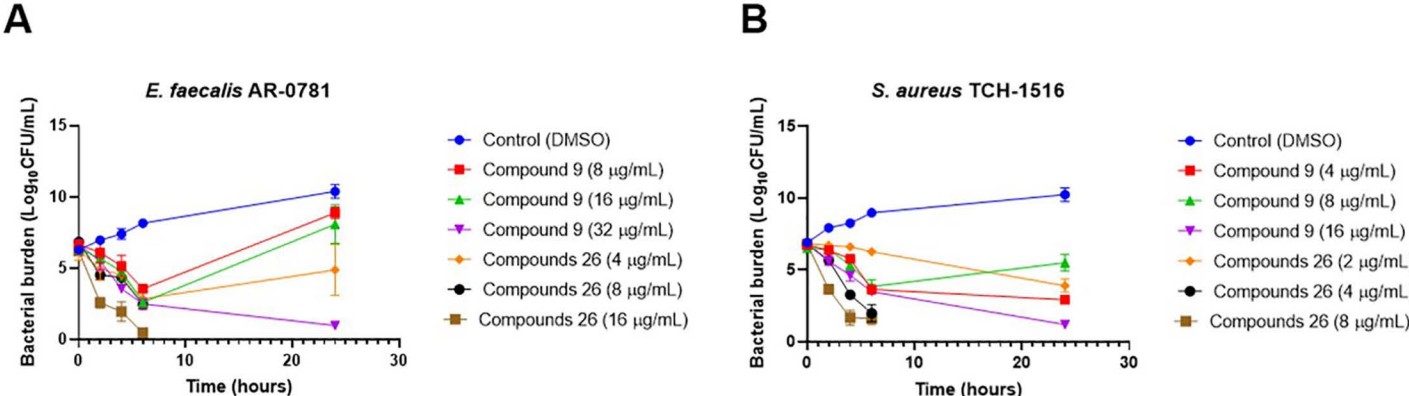

**Fig 3. The time-kill kinetics of compounds 9 and 26 against multidrug-resistant *E. faecalis* AR-0781 (panel A) and *S. aureus* TCH-1516 (panel B) isolates.** The bacterial strains were exposed to sub-MIC, MIC, and 1×MIC concentrations of each compound and incubated for 24 hours. At designated time points, the microbial cultures were aliquoted, serially diluted, and plated on sheep blood agar plates. The colonies were counted, and bacterial burden (expressed as log₁₀ CFU/mL) was calculated. The data are presented as the mean±SD of three experimental replicates.

## Discussion

In this study we describe the synthesis of novel *N*-substituted *β*-amino acid derivatives bearing 2-hydroxyphenyl moieties as promising antimicrobial candidates targeting drug-resistant Gram-positive priority pathogens with genetically defined resistance mechanisms. This study identifies a promising *N*-substituted *β*-amino acid-based scaffold with broad Gram-positive pathogens-directed activity for further hit to lead optimization.

The ESKAPE group pathogens, comprising *Enterococcus faecium, Staphylococcus aureus, Klebsiella pneumoniae*, *Acinetobacter baumannii*, *Pseudomonas aeruginosa*, and *Enterobacter* species, are extremely challenging for their role in hospital-acquired infections and multidrug resistance [2,4,13]. Among these, vancomycin-resistant *Enterococcus* (VRE), including *E. faecalis* as well as methicillin-resistant *Staphylococcus aureus* (MRSA) pose significant clinical challenges due to their extensive resistance to multiple antibiotics, complicating therapeutic strategies and leading to fatal outcomes [10,13,30]. Epidemiologically, VRE and MRSA are highly prevalent in nosocomial environments, contributing to severe infections, increased morbidity and mortality rates, and substantial healthcare expenditures [30]. The persistent difficulties in managing infections caused by these pathogens highlight the critical need for the development of novel antimicrobial agents.

*N*-substituted *β*-amino acid derivatives bearing phenolic groups represent a promising class of compounds due to their potential as pharmacophores [25,26,32]. These derivatives exhibit strong hydrogen-bonding capabilities and the ability to potentially form stable interactions with microbial cellular targets, thereby leading to the antimicrobial activity [25,26,32]. The incorporation of phenolic groups confers additional bactericidal properties, while aromatic or heterocyclic substitutions can further augment the binding affinity and specificity of these molecules [25,28,30]. In this study, we successfully synthesized a series of *N*-substituted *β*-amino acid derivatives bearing a 2-hydroxyphenyl group. These compounds were designed to represent a range of aromatic and heterocyclic substituents and exhibited a structure-dependent antimicrobial activity, predominantly against multidrug-resistant Gram-positive pathogens. Among the synthesized compounds, compounds **9** (R = 4-nitrophenyl), **17** (R = 5-nitro-2-thienyl), **18** (R = 5-nitro-2-furyl), thiosemicarbazide **16**, and **26** demonstrated the most potent activity against *S. aureus* MRSA USA300 strain TCH-1516, with minimum inhibitory concentration values ranging from 4 to 16 μg/mL. The enhanced antimicrobial activity observed for these compounds can be attributed to the physicochemical properties of the substituents as well as geenral *N*-substituted *β*-amino acid or 2-hydroxyphenyl

pharmacophores. The nitro groups present in compounds **9, 17**, and **18** are strong electron-withdrawing moieties, which increase the electrophilicity of the aromatic ring and may enhance the binding affinity to various bacterial target sites. Furthermore, the heterocyclic rings in compounds **17** (thienyl) and **18** (furyl) provide additional molecular properties for interaction through π-π stacking and hydrogen bonding with bacterial enzymes or membrane components. Compound **26**, bearing a thiosemicarbazide group, demonstrated significant antimicrobial activity against vancomycin-resistant *E. faecalis* AR-0781, with an activity profile comparable to that of control antibiotics.

The thiosemicarbazide moiety, containing sulfur and nitrogen atoms, is potentially involved in the formation of strong intermolecular interactions, including hydrogen bonding and coordination with metal ions, which may contribute to its potent bioactivity. In our previous studies, we successfully synthesized *N*-substituted *β*-amino acid derivatives bearing a 3-hydroxyphenyl core, which exhibited notable antimicrobial activity against Gram-positive drug-resistant pathogens as well as drug-resistant *Candida* species [28]. These findings suggest that the *N*-substituted *β*-amino acid scaffold represents a promising platform for the discovery of novel antimicrobial agents. Importantly, the position of the hydroxyl group on the phenolic ring appears to play a critical role in modulating the antimicrobial spectrum [33,34]. Specifically, phenolic groups with hydroxyl substitutions at different positions on the aromatic ring may influence both antibacterial and antifungal activities. Substitutions in the meta-position, as seen in our previous derivatives, seem to enhance activity against Gram-positive bacteria, while the incorporation of hydroxyl groups at other positions could potentially broaden the spectrum to include fungal pathogens. The data presented in this study underscore the significant potential of *N*-substituted *β*-amino acid derivatives as novel antimicrobial agents, particularly due to their selective activity against multidrug-resistant Gram-positive pathogens. The structure-activity relationship observed, especially with regard to the position of the phenolic hydroxyl group and the nature of the substituents, highlights the critical role of molecular design in optimizing antimicrobial efficacy. The strong activity exhibited by compounds containing nitro-aromatic and heterocyclic moieties reinforces the importance of electron-withdrawing groups and π-stacking interactions in effectively targeting resistant Gram-positive bacterial strains. The broad-spectrum activity of compound **26** against both vancomycin-resistant *Enterococcus faecalis* and methicillin-resistant *Staphylococcus aureus* suggests that this scaffold could serve as a foundation for further optimization to address other high-priority pathogens, including those identified by the World Health Organization as critical threats. However, despite the promising results, this study has several limitations. First, additional screening is needed against a broader range of bacterial strains with diverse drug-resistant phenotypes to fully assess the efficacy of the most promising compounds. Second, although strong activity was observed against *S. aureus* and *E. faecalis*, there remains a gap in our understanding of how these compounds perform against other clinically significant Gram-positive pathogens, such as *Streptococcus* species or other *Staphylococcus* strains with other resistance and virulence mechanisms.

Finally, further studies are required to elucidate the specific bacterial targets of these *N*-substituted *β*-amino acid derivatives, which will be crucial for further advancing their development as therapeutic candidates for the early pre-clinical screening.

## Conclusions

In this study, we report the synthetic pathways for the generation of novel libraries of 3,3'-((2-hydroxyphenyl)azanediyl) dipro-pionic acid derivatives, incorporating functional groups such as ester, hydrazine, hydrazones, benzimidazole, dimethylpyrrole, dimethylpyrazole, and oxadezipine moieties. These compounds demonstrated promising structure-depended antimicrobial activity against multidrug-resistant Gram-positive pathogens, notably methicillin-resistant *Staphylococcus aureus* (MRSA) and vancomycin-resistant *Enterococcus* (VRE). Structure-activity relationship (SAR) analysis revealed that electron-withdrawing nitro-aromatic and heterocyclic substituents significantly enhanced antimicrobial efficacy. Compound **26** exhibited broad-spectrum antibacterial and antifungal activity, demonstrating the potential of this scaffold for further hit-to-lead optimization. While these initial results are encouraging, further studies are needed to understand the activity of

compound **26** and various compound **26**-based derivatives against a broader range of drug-resistant Gram-positive isolates. These findings highlights that *N*-substituted *β*-amino acid derivatives could be explored as a promising platform for the development of novel antimicrobial agents targeting predominantly multidrug-resistant Gram-positive pathogens.

## Supporting information

**S1 File.  Figure S1–S49.** $^1$H and $^{13}$C NMR spectra of compounds **2−26** (in DMSO-$d_6$). Table S1. The *in vitro* antimicrobial activity of *N*-substituted *β*-amino acid derivatives **2–26** against panel of fungal pathogens.
(DOCX)

## Acknowledgments

We are thankful for the supportive staff of Weill Cornell Medicine of Cornell University and Kaunas University of Technology for their immense technical help and support during this study.

## Author contributions

**Conceptualization:** Povilas Kavaliauskas, Birutė Grybaitė, Vytautas Mickevičius, Vidmantas Petraitis.

**Data curation:** Povilas Kavaliauskas, Birutė Grybaitė, Vytautas Mickevičius, Vidmantas Petraitis.

**Formal analysis:** Povilas Kavaliauskas, Birutė Grybaitė, Birute Sapijanskaite-Banevič, Rūta Petraitienė, Andrew Garcia, Ethan Naing, Vytautas Mickevičius, Vidmantas Petraitis.

**Funding acquisition:** Vidmantas Petraitis.

**Investigation:** Povilas Kavaliauskas, Birutė Grybaitė, Birute Sapijanskaite-Banevič, Rūta Petraitienė, Ramunė Grigalevičiūtė, Andrew Garcia, Ethan Naing, Vytautas Mickevičius, Sergey Belyakov, Vidmantas Petraitis.

**Methodology:** Povilas Kavaliauskas, Birutė Grybaitė, Birute Sapijanskaite-Banevič, Vytautas Mickevičius.

**Supervision:** Povilas Kavaliauskas.

**Visualization:** Povilas Kavaliauskas.

**Writing – original draft:** Povilas Kavaliauskas, Birutė Grybaitė, Vytautas Mickevičius, Vidmantas Petraitis.

**Writing – review & editing:** Povilas Kavaliauskas, Birutė Grybaitė, Vytautas Mickevičius, Vidmantas Petraitis.

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
