## [Decision Letter · Decision Letter 0]

11 Oct 2024

PONE-D-24-42309Synthesis of Novel N-Substituted β-Amino Acid Derivatives Bearing 2-Hydroxyphenyl Moieties as Promising Antimicrobial Candidates Targeting Multidrug-Resistant Gram-Positive PathogensPLOS ONE

Dear Dr. Kavaliauskas,

Thank you for submitting your manuscript to PLOS ONE. After careful consideration, we feel that it has merit but does not fully meet PLOS ONE’s publication criteria as it currently stands. Therefore, we invite you to submit a revised version of the manuscript that addresses the points raised during the review process.

We look forward to receiving your revised manuscript.

Kind regards,

Wagdy M. Eldehna, Ph.d

Academic Editor

PLOS ONE

Journal Requirements:

2. We note that this submission includes NMR spectroscopy data. We would recommend that you include the following information in your methods section or as Supporting Information files:

a) The make/source of the NMR instrument used in your study, as well as the magnetic field strength. For each individual experiment, please also list: the nucleus being measured; the sample concentration; the solvent in which the sample is dissolved and if solvent signal suppression was used; the reference standard and the temperature.

b) A list of the chemical shifts for all compounds characterised by NMR spectroscopy, specifying, where relevant: the chemical shift (δ), the multiplicity and the coupling constants (in Hz), for the appropriate nuclei used for assignment.

c)The full integrated NMR spectrum, clearly labelled with the compound name and chemical structure.

We also strongly encourage authors to provide primary NMR data files, in particular for new compounds which have not been characterised in the existing literature. Authors should provide the acquisition data, FID files and processing parameters for each experiment, clearly labelled with the compound name and identifier, as well as a structure file for each provided dataset. See our list of recommended repositories here: https://journals.plos.org/plosone/s/recommended-repositories .

Reviewers' comments:

Reviewer's Responses to Questions

**Comments to the Author**

1. Is the manuscript technically sound, and do the data support the conclusions?

Reviewer #1: Partly

Reviewer #2: Yes

Reviewer #3: Yes

Reviewer #4: Yes

2. Has the statistical analysis been performed appropriately and rigorously? 

Reviewer #1: N/A

Reviewer #2: Yes

Reviewer #3: Yes

Reviewer #4: Yes

3. Have the authors made all data underlying the findings in their manuscript fully available?

Reviewer #1: Yes

Reviewer #2: Yes

Reviewer #3: Yes

Reviewer #4: Yes

4. Is the manuscript presented in an intelligible fashion and written in standard English?

Reviewer #1: Yes

Reviewer #2: Yes

Reviewer #3: Yes

Reviewer #4: Yes

5. Review Comments to the Author

Reviewer #1: Review “ Synthesis of Novel N-Substituted β-Amino Acid Derivatives Bearing 2-Hydroxyphenyl Moieties as Promising Antimicrobial Candidates Targeting Multidrug-Resistant Gram-Positive Pathogens“

Lines 51-53 Check citations. Enterococcus species or Enterococcus faecium? It is important.

Line 57 Check Enterococcus or Enterococcci or Enterococcus faecium. What do you mean “Enterococcus“?

Line 441 Check the name E. clocea or E. cloacae ? Write the full name of this pathogen when you mention it for the first time.

Line 442 Institute of Infectious Diseases and Pathogenic Microbiology collection. Where is it?

Line 446 What about fungal strains? It must be described.

Line 453 A producer of 96-well microplates is not mentioned.

Line 455 A producer of CAMHB is not mentioned.

Line 456 A producer of RMPI/MOPS media is not mentioned.

Line 463 “tested drug“ or tested compound?

Line 466 The control of antibacterial and antifungal agents is not described. The producers of antibiotics must be mentioned.

Line 521 You must write Enterococcus faecalis. Do not us Italic for spp. !

Line 522 spp. but not spp

Line 522 E. cloacae but not Enterobacter spp

Line 530 You used bacterial strains and isolates, but not only isolates. The title of the table 1 must be improved.

Table 1 Explain, what “128>“ means. You write “128>“ when the compound concentration of 128 mg/mL showed no antimicrobial activity and the higher concentration equal to 256 mg/mL was not tested. It would be better to write that the results were negative. “128 >“ is an interpretation of your results? In my opinion, the presentation of data in the table 1 is subject to discussion or correction. Furthermore, in the abstract, you write that “one of the compounds demonstrated activity against Gram-negative pathogens“.

Explain why MBCs of compounds were not determined.

Line 614 “Enterococcus species.,“? Check grammar! Isn‘t the member of ESKAPE group only Enterococcus faecium? Is E. faecalis also a member of ESKAPE group?

Line 678 Staphylococcus but not Staphylococcus

Line 684 “Collectively, in this study,“ The beginning of conclusions does not seem good.

Use Italic when it is required as writen in an attachent.

Reviewer #2: This manuscript is devoted to the development of new antimicrobial drugs. The relevance of this issue is undeniable and constant, since it is necessary to continuously update the pool of these drugs due to the resistance to active substances developed by pathogenic organisms. The authors focused their attention on N-substituted β-amino acid derivatives bearing a 2-hydroxyphenyl moiety, synthesizing a number of compounds containing this fragment and belonging to several very different classes of organic compounds, including heterocyclic ones. Having determined their activity in relation to several types of pathogenic microorganisms, they found that the substituted 1,3,4-oxadiazole-2(3H)-thione showed the greatest activity.

In this regard, I have a wish for the authors: it is worth discussing in more detail which fragment determines the activity in this case - β-amino acid or 1,3,4-oxadiazole-2(3H)-thione.

From a chemical point of view, the work is of particular interest in that the authors did not limit themselves to one or two reactions, but demonstrated the richness of organic synthesis and the diversity of chemical properties of carboxylic acids. Personally, I found it interesting to read the manuscript. The experiment was performed well. I have no doubts about the structure of the obtained compounds, since they are almost completely characterized, and the quality of the presented NMR spectra is quite high.

Several comments on the description of the results are given below:

1. In Scheme 3 and in the text that describes it, the initial dicarbonyl compounds (acetylacetone and acetonylacetone) are confused.

2. It would be very good if the authors could explain why in the 1H NMR spectra of dihydrazides 7-14 there are 3 signals of the phenol proton and 3-4 signals of the NH protons, while in the 13C NMR spectra of some of these compounds there are 2 signals of carbonyl carbon atoms. There are two questions here: a) why are there several signals at all, despite the apparent symmetry of these compounds; b) why the number of forms according to 1H and 13C NMR data is sometimes different? It is desirable to add a discussion of this fact to the discussion section devoted to these compounds.

3. In the description of the fluorine-substituted derivative (compound 8), the 19F NMR spectrum should be added, and either the signals of carbon atoms with C-F coupling constants should be indicated, or the spectrum with fluorine decoupling should be registered. Currently, the 13C NMR spectrum does not formally correspond to the proposed structure, since it contains 29 signals of aromatic protons, which exceeds all reasonable limits.

4. The description of the synthetic methods contains many typos and language errors. For example, in the description of the synthesis of compound 3, for some reason, the present tense is used instead of the past tense, as in the description of compound 2 (and as it should be). Moreover, this is far from the only example of a tense discrepancy, sometimes even within a single sentence. Examples of other inaccuracies (not exhaustive):

- Line 138. "the reaction mixture 20 minutes stirred" should be changed to "the reaction mixture was stirred for 20 minutes".

- Line 154. "the residuse neutralized" should be changed to "the residue was neutralized".

The authors should proofread this section more carefully.

Reviewer #3: - The study has been well designed, organized, worked and prepared. The topic is important and actual. It seems the continuation of the previous work (ref.28) in order to investigate and compare the effects of the (various) substituent groups in different positions on the drug-resistant patogens.

- In the results part, H-NMR values are in 2 decimals, but also C-NMR values are in 2 decimals. However, the C-NMR values should be written in 1 decimal in a scientific paper.

- In some H-NMR and C-NMR results, the numbers of protons and carbons are inconsistent with the spectra/structures, this should be checked.

- There are some additional/extra peaks for the H-NMR and C-NMR results, are they impurities? All purifications in the work were performed by crystallization/crystals and any other methods (e.g. column chromatography) were apply to purify the compounds, was this enough for a good purification? Because, especially for the biological activity tests purity/impurities are of high importance.

- Mass spectroscopy analyses are present for some compounds, not for all, is there a specific reason for that?

- Are the synthesized compounds new? If yes, this was not really emphasized in the manuscript and abstract, this is important and should be mentioned in the appropriate parts of the manuscript. If no, then the related references must be given.

- The preparation purposes of nearly all compound were well explained in the synthesis part except compound-6. It should also be mentioned in this part.

- Between the lines 582 and 603, the same issue was discussed and repeated twice, this should be checked and rearranged.

- Line 693: resistant; twice.

Reviewer #4: The article entitled “Synthesis of Novel N-Substituted β-Amino Acid Derivatives Bearing 2-Hydroxyphenyl Moieties as Promising Antimicrobial Candidates Targeting Multidrug-Resistant Gram-Positive Pathogens” describes the synthesis and biological evaluation of N-Substituted β-Amino Acid Derivatives with a 2-hydroxyphenyl moeity as the antimicrobial agents. The manuscript would be of general interest to the researchers of this field. Before recommending for publication, the authors should consider and incorporate in the present form of the manuscript of some concerns and comments that need to be addressed.

Some comments and corrections for authors:

1. Overally, the manuscript has some punctuation and grammatical errors and needs to be corrected (i.e., there must be comma before and in all mns when mention about over two parameters). Please run throughout the mns.

2. A figure showing the importance of N-substitution in β-amino acids in med chem from literature should be added to the intro section.

3. The reaction pathway is clear. But some of compounds were not fully characterized. Some molecules do not have FTIR analysis, others do not have HRMS analysis.

4. The authors must define the unknow/known compounds in the mns.

5. I’d prefer to see the X-ray single analyses of unexpected products such as 3, 6, and 24.

6. “ml” must be corrected as “mL”. Please check and correct throughout the mns.

7. The data of carbon NMR values should be one decimal after point.

8. I strongly recommend to the authors to support their findings with in silico methods and further studies such as time-kill assay and/or biofilm formation especially for 9 and 26.

6. PLOS authors have the option to publish the peer review history of their article (what does this mean? ). If published, this will include your full peer review and any attached files.

**Do you want your identity to be public for this peer review?** For information about this choice, including consent withdrawal, please see our Privacy Policy .

Reviewer #1: No

Reviewer #2: No

Reviewer #3: No

Reviewer #4: No

---

## [Author Response · Author response to Decision Letter 1]

31 Oct 2024

Please find the response bellow:

PONE-D-24-42309R1

Synthesis of Novel N-Substituted β-Amino Acid Derivatives Bearing 2-Hydroxyphenyl Moieties as Promising Antimicrobial Candidates Targeting Multidrug-Resistant Gram-Positive Pathogens

Dr. Povilas Kavaliauskas

Dear Dr. Kavaliauskas,

We've checked your submission and before we can proceed, we need you to address the following issues:

1. We notice that your manuscript file was uploaded on September 23, 2024. Please can you upload the latest version of your revised manuscript as the main article file, ensuring that does not contain any tracked changes or highlighting. This will be used in the production process if your manuscript is accepted. Please follow this link for more information: http://blogs.PLOS.org/everyone/2011/05/10/how-to-submit-your-revised-manuscript/

We have uploaded the latest version of the manuscript without any tracs.

2. We note that this submission includes NMR spectroscopy data. We would recommend that you include the following information in your methods section or as Supporting Information files:

a) The make/source of the NMR instrument used in your study, as well as the magnetic field strength. For each individual experiment, please also list: the nucleus being measured; the sample concentration; the solvent in which the sample is dissolved and if solvent signal suppression was used; the reference standard and the temperature.

We would like to emphasize that this information is included in the manuscript.

THe description of the instruments used is indicated at L 117-124.

b) A list of the chemical shifts for all compounds characterised by NMR spectroscopy, specifying, where relevant: the chemical shift (δ), the multiplicity and the coupling constants (in Hz), for the appropriate nuclei used for assignment.

The requested information is provided at L126-440.

c)The full integrated NMR spectrum, clearly labelled with the compound name and chemical structure.

We also strongly encourage authors to provide primary NMR data files, in particular for new compounds which have not been characterised in the existing literature. Authors should provide the acquisition data, FID files and processing parameters for each experiment, clearly labelled with the compound name and identifier, as well as a structure file for each provided dataset. See our list of recommended repositories here: https://journals.plos.org/plosone/s/recommended-repositories.

The NMR spectra are provided in the supplementary file. They include structures, information about solvent and full uncropped spectrum.

Upon discussion with all authors, we believed that the NMR spectrum provided in the manuscript is sufficient to show the identity of the compounds, there we collectively decided not to proceed with the spectra repositories.

---

## [Decision Letter · Decision Letter 1]

26 Nov 2024

PONE-D-24-42309R1Synthesis of Novel N-Substituted β-Amino Acid Derivatives Bearing 2-Hydroxyphenyl Moieties as Promising Antimicrobial Candidates Targeting Multidrug-Resistant Gram-Positive PathogensPLOS ONE

Dear Dr. Kavaliauskas,

Thank you for submitting your manuscript to PLOS ONE. After careful consideration, we feel that it has merit but does not fully meet PLOS ONE’s publication criteria as it currently stands. Therefore, we invite you to submit a revised version of the manuscript that addresses the points raised during the review process.

We look forward to receiving your revised manuscript.

Kind regards,

Wagdy M. Eldehna, Ph.d

Academic Editor

PLOS ONE

Additional Editor Comments:

Dear Authors,

I hope this message finds you well. Upon reviewing your recent submission, I noticed that you have addressed the comments from only one reviewer. This may have been an oversight, and I want to ensure that all feedback is considered.

To proceed, please upload your responses to all reviewers' comments. If you do not provide a comprehensive response, I regret to inform you that the paper will be at risk of rejection.

Thank you for your attention to this matter.

Reviewers' comments:

Reviewer's Responses to Questions

**Comments to the Author**

1. If the authors have adequately addressed your comments raised in a previous round of review and you feel that this manuscript is now acceptable for publication, you may indicate that here to bypass the “Comments to the Author” section, enter your conflict of interest statement in the “Confidential to Editor” section, and submit your "Accept" recommendation.

Reviewer #2: (No Response)

Reviewer #4: (No Response)

2. Is the manuscript technically sound, and do the data support the conclusions?

Reviewer #2: No

Reviewer #4: Yes

3. Has the statistical analysis been performed appropriately and rigorously? 

Reviewer #2: (No Response)

Reviewer #4: Yes

4. Have the authors made all data underlying the findings in their manuscript fully available?

Reviewer #2: (No Response)

Reviewer #4: Yes

5. Is the manuscript presented in an intelligible fashion and written in standard English?

Reviewer #2: (No Response)

Reviewer #4: Yes

6. Review Comments to the Author

Reviewer #2: None of my comments were taken into account. Therefore, I recommend that this manuscript be rejected.

Reviewer #4: I could not see the "rebuttal letter" for my comments and recommendations. In this case, I am sorry that the mns should be rejected because authors have not addressed my comments.

7. PLOS authors have the option to publish the peer review history of their article (what does this mean? ). If published, this will include your full peer review and any attached files.

**Do you want your identity to be public for this peer review?** For information about this choice, including consent withdrawal, please see our Privacy Policy .

Reviewer #2: No

Reviewer #4: No

---

## [Author Response · Author response to Decision Letter 2]

26 Feb 2025

To the Editor of PLOS One,

To the reviewers

Dear Sir/Madam,

On behalf of all authors I would like to thank the reviewers for their expertise and valuable comments aimed to improve this manuscript.

We have responded to all comments and incorporated the changes to the manuscript.

Bellow please find the detailed response to the comments.

We hope that the current, updated form of the manuscript will meet the merit of PLOS One journal.

Sincerely,

Povilas Kavaliauskas

Reviewer #2:

This manuscript is devoted to the development of new antimicrobial drugs. The relevance of this issue is undeniable and constant, since it is necessary to continuously update the pool of these drugs due to the resistance to active substances developed by pathogenic organisms. The authors focused their attention on N-substituted β-amino acid derivatives bearing a 2-hydroxyphenyl moiety, synthesizing a number of compounds containing this fragment and belonging to several very different classes of organic compounds, including heterocyclic ones. Having determined their activity in relation to several types of pathogenic microorganisms, they found that the substituted 1,3,4-oxadiazole-2(3H)-thione showed the greatest activity.

In this regard, I have a wish for the authors: it is worth discussing in more detail which fragment determines the activity in this case - β-amino acid or 1,3,4-oxadiazole-2(3H)-thione.

From a chemical point of view, the work is of particular interest in that the authors did not limit themselves to one or two reactions, but demonstrated the richness of organic synthesis and the diversity of chemical properties of carboxylic acids. Personally, I found it interesting to read the manuscript. The experiment was performed well. I have no doubts about the structure of the obtained compounds, since they are almost completely characterized, and the quality of the presented NMR spectra is quite high.

Several comments on the description of the results are given below:

1. In Scheme 3 and in the text that describes it, the initial dicarbonyl compounds (acetylacetone and acetonylacetone) are confused.

We are thankful for the comment. We have corrected it.

2. It would be very good if the authors could explain why in the 1H NMR spectra of dihydrazides 7-14 there are 3 signals of the phenol proton and 3-4 signals of the NH protons, while in the 13C NMR spectra of some of these compounds there are 2 signals of carbonyl carbon atoms.

We are thankful for the comment. This is related the restricted rotation around the CONH led to the formation in an isomeric mixture of hydrazones where Z isomer predominates. The obtained hydrazones 7−14 show double sets of resonances for the N=CH and CONH fragment protons with the intensity ratio of 0.3:0.7 (1H NMR). No formation of geometrical isomers was observed.

There are two questions here: a) why are there several signals at all, despite the apparent symmetry of these compounds;

We are thankful for the comment. As we mentioned before. This is related the restricted rotation around the CONH led to the formation in an isomeric mixture of hydrazones where Z isomer predominates. The obtained hydrazones 7−14 show double sets of resonances for the N=CH and CONH fragment protons with the intensity ratio of 0.3:0.7 (1H NMR). No formation of geometrical isomers was observed.

b) why the number of forms according to 1H and 13C NMR data is sometimes different? It is desirable to add a discussion of this fact to the discussion section devoted to these compounds.

We are thankful for the comment. We have added this point in the discussion section.

3. In the description of the fluorine-substituted derivative (compound 8), the 19F NMR spectrum should be added, and either the signals of carbon atoms with C-F coupling constants should be indicated, or the spectrum with fluorine decoupling should be registered. Currently, the 13C NMR spectrum does not formally correspond to the proposed structure, since it contains 29 signals of aromatic protons, which exceeds all reasonable limits.

We are thankful for the comment. We have performed the additional experiments, where we recorded 19F NMR. THe results are added 19F NMR in the compound 8 experimental part description.

4. The description of the synthetic methods contains many typos and language errors. For example, in the description of the synthesis of compound 3, for some reason, the present tense is used instead of the past tense, as in the description of compound 2 (and as it should be). Moreover, this is far from the only example of a tense discrepancy, sometimes even within a single sentence. Examples of other inaccuracies (not exhaustive):

We are thankful for the comment. We have corrected it.

- Line 138. "the reaction mixture 20 minutes stirred" should be changed to "the reaction mixture was stirred for 20 minutes".

We are thankful for the comment. We have corrected it.

- Line 154. "the residuse neutralized" should be changed to "the residue was neutralized".

The authors should proofread this section more carefully.

We are thankful for the comment. We have corrected it.

Reviewer #3:

- The study has been well designed, organized, worked and prepared. The topic is important and actual. It seems the continuation of the previous work (ref.28) in order to investigate and compare the effects of the (various) substituent groups in different positions on the drug-resistant patogens.

- In the results part, H-NMR values are in 2 decimals, but also C-NMR values are in 2 decimals. However, the C-NMR values should be written in 1 decimal in a scientific paper.

We are thankful for the comment. We have corrected it.

- In some H-NMR and C-NMR results, the numbers of protons and carbons are inconsistent with the spectra/structures, this should be checked.

We are thankful for the comment. We have have checked and corrected it.

- There are some additional/extra peaks for the H-NMR and C-NMR results, are they impurities? All purifications in the work were performed by crystallization/crystals and any other methods (e.g. column chromatography) were apply to purify the compounds, was this enough for a good purification? Because, especially for the biological activity tests purity/impurities are of high importance.

We agree with the reviewers comment regarding potential impurities or break down products. Based on your available purification methods, we aimed to obtain compounds as pure as possible. With that being said, we take this as a potential limitation of this study. With methods available to our group, we obtained compounds with reasonable purity of compounds, as confirmed by NMR, elemental analysis, and mass spectrometry data.

- Mass spectroscopy analyses are present for some compounds, not for all, is there a specific reason for that?

Due to the institutional changes as well as maintenance problems, the mass spectrometer is no longer available to us and our collaborators. Therefore, we are unable to record this data. We hope that NMR spectrums and other supporting constants will be sufficient to prove the identityof the synthesized compounds.

- Are the synthesized compounds new? If yes, this was not really emphasized in the manuscript and abstract, this is important and should be mentioned in the appropriate parts of the manuscript. If no, then the related references must be given.

We are thankful for the comment. Yes, all synthesized compounds are new, it were mentioned in the corresponding parts of the manuscript.

- The preparation purposes of nearly all compound were well explained in the synthesis part except compound-6. It should also be mentioned in this part.

We are thankful for the comment. We have corrected it.

- Between the lines 582 and 603, the same issue was discussed and repeated twice, this should be checked and rearranged.

- Line 693: resistant; twice.

We have addressed this comment and incorporated the changes.

Reviewer #4:

The article entitled “Synthesis of Novel N-Substituted β-Amino Acid Derivatives Bearing 2-Hydroxyphenyl Moieties as Promising Antimicrobial Candidates Targeting Multidrug-Resistant Gram-Positive Pathogens” describes the synthesis and biological evaluation of N-Substituted β-Amino Acid Derivatives with a 2-hydroxyphenyl moeity as the antimicrobial agents. The manuscript would be of general interest to the researchers of this field. Before recommending for publication, the authors should consider and incorporate in the present form of the manuscript of some concerns and comments that need to be addressed.

Some comments and corrections for authors:

1. Overally, the manuscript has some punctuation and grammatical errors and needs to be corrected (i.e., there must be comma before and in all mns when mention about over two parameters). Please run throughout the mns.

We are thankful for the comment. We have corrected it.

2. A figure showing the importance of N-substitution in β-amino acids in med chem from literature should be added to the intro section.

We are thankful for the comment. We have corrected it.

Figure 1. Pharmaceuticals containing β-amino acid structures.

3. The reaction pathway is clear. But some of compounds were not fully characterized. Some molecules do not have FTIR analysis, others do not have HRMS analysis.

We are thankful for the comment. But we disagree with the first statement, we have the FTIR analysis for all synthesized compounds. Due to institutional challenges and the broken instrument, we are not able to obtain HRMS data.

4. The authors must define the unknow/known compounds in the mns.

We are thankful for the comment. We have corrected it.

5. I’d prefer to see the X-ray single analyses of unexpected products such as 3, 6, and 24.

We are thankful for the comment. To address this, we have performed X ray cristalography of compound 24. We are providing this data as Figure 1.

Figure 1. The ORTEP diagram of analysed compound 24 showing the numbering scheme used in this study.

The analytical data is provided as a supplement and can be found as:

Table S2. Crystal data and structure refinement for compound 24 (named BGiii159).

Table S3. Fractional Atomic Coordinates (×104) and Equivalent Isotropic Displacement Parameters (Å2×103) for compound 24 (named BGiii159). Ueq is defined as 1/3 of of the trace of the orthogonalised UIJ tensor.

Table S4. Anisotropic Displacement Parameters (Å2×103) for compound 24 (named BGiii159). The Anisotropic displacement factor exponent takes the form: -2π2[h2a*2U11+2hka*b*U12+…].

Table S5. Bond Lengths for compound 24 (Named BGiii159).

Table S6. Bond Angles for compound 24 (Named BGiii159).

Table S7. Torsion Angles for compound 24 (Named BGiii159).

Table S8. Hydrogen Atom Coordinates (Å×104) and Isotropic Displacement Parameters (Å2×103) for compound 24 (Named BGiii159).

6. “ml” must be corrected as “mL”. Please check and correct throughout the mns.

We have incorporated the changes.

7. The data of carbon NMR values should be one decimal after point.

We have incorporated the changes.

8. I strongly recommend to the authors to support their findings with in silico methods and further studies such as time-kill assay and/or biofilm formation especially for 9 and 26.

We are thankful for the reviewer for their great suggestion. As requested, we performed time-kill assay t propose the mechanisms of action. We are introducing this data to figure 2.

Figure 2. The time-kill kinetics of compounds 9 and 26 against multidrug-resistant E. faecalis AR-0781 (panel A) and S. aureus TCH-1516 (panel B) isolates. The bacterial strains were exposed to sub-MIC, MIC, and 1× MIC concentrations of each compound and incubated for 24 hours. At designated time points, the microbial cultures were aliquoted, serially diluted, and plated on sheep blood agar plates. The colonies were counted, and bacterial burden (expressed as log₁₀ CFU/mL) was calculated. The data are presented as the mean ± SD of three experimental replicates.

---

## [Decision Letter · Decision Letter 2]

7 Mar 2025

Synthesis of Novel N-Substituted β-Amino Acid Derivatives Bearing 2-Hydroxyphenyl Moieties as Promising Antimicrobial Candidates Targeting Multidrug-Resistant Gram-Positive Pathogens

PONE-D-24-42309R2

Dear Dr. Kavaliauskas,

We’re pleased to inform you that your manuscript has been judged scientifically suitable for publication and will be formally accepted for publication once it meets all outstanding technical requirements.

Kind regards,

Wagdy M. Eldehna, Ph.d

Academic Editor

PLOS ONE

Additional Editor Comments (optional):

Reviewers' comments:

Reviewer's Responses to Questions

**Comments to the Author**

1. If the authors have adequately addressed your comments raised in a previous round of review and you feel that this manuscript is now acceptable for publication, you may indicate that here to bypass the “Comments to the Author” section, enter your conflict of interest statement in the “Confidential to Editor” section, and submit your "Accept" recommendation.

Reviewer #2: All comments have been addressed

Reviewer #4: All comments have been addressed

2. Is the manuscript technically sound, and do the data support the conclusions?

Reviewer #2: Yes

Reviewer #4: Yes

3. Has the statistical analysis been performed appropriately and rigorously? 

Reviewer #2: Yes

Reviewer #4: Yes

4. Have the authors made all data underlying the findings in their manuscript fully available?

Reviewer #2: (No Response)

Reviewer #4: Yes

5. Is the manuscript presented in an intelligible fashion and written in standard English?

Reviewer #2: Yes

Reviewer #4: Yes

6. Review Comments to the Author

Reviewer #2: (No Response)

Reviewer #4: The study has been well designed, organized, and well prepared according to the reviewers' comments. The manuscript can be accepted.

7. PLOS authors have the option to publish the peer review history of their article (what does this mean? ). If published, this will include your full peer review and any attached files.

**Do you want your identity to be public for this peer review?** For information about this choice, including consent withdrawal, please see our Privacy Policy .

Reviewer #2: No

Reviewer #4: No

---

## [Editor Report · Acceptance letter]

PONE-D-24-42309R2

PLOS ONE

Dear Dr. Kavaliauskas,

I'm pleased to inform you that your manuscript has been deemed suitable for publication in PLOS ONE. Congratulations! Your manuscript is now being handed over to our production team.

Kind regards,

on behalf of

Dr. Wagdy M. Eldehna

Academic Editor

PLOS ONE